# Cilia structure and intraflagellar transport differentially regulate sensory response dynamics within and between *C. elegans* chemosensory neurons

**Alison Philbrook**[1]*, **Michael P. O'Donnell**[2], **Laura Grunenkovaite**[1], **Piali Sengupta**[1]*

1 Department of Biology, Brandeis University, Waltham, Massachusetts, United States of America,
2 Department of Molecular, Cellular, and Developmental Biology, Yale University, Connecticut, United States of America

* aphilbrook@brandeis.edu (AP); sengupta@brandeis.edu (PS)

## Abstract

Sensory neurons contain morphologically diverse primary cilia that are built by intraflagellar transport (IFT) and house sensory signaling molecules. Since both ciliary structural and signaling proteins are trafficked via IFT, it has been challenging to decouple the contributions of IFT and cilia structure to neuronal responses. By acutely inhibiting IFT without altering cilia structure and vice versa, here we describe the differential roles of ciliary trafficking and sensory ending morphology in shaping chemosensory responses in *Caenorhabditis elegans*. We show that a minimum cilium length but not continuous IFT is necessary for a subset of responses in the ASH nociceptive neurons. In contrast, neither cilia nor continuous IFT are necessary for odorant responses in the AWA olfactory neurons. Instead, continuous IFT differentially modulates response dynamics in AWA. Upon acute inhibition of IFT, cilia-destined odorant receptors are shunted to ectopic branches emanating from the AWA cilia base. Spatial segregation of receptors in these branches from a cilia-restricted regulatory kinase results in odorant desensitization defects, highlighting the importance of precise organization of signaling molecules at sensory endings in regulating response dynamics. We also find that adaptation of AWA responses upon repeated exposure to an odorant is mediated by IFT-driven removal of its cognate receptor, whereas adaptation to a second odorant is regulated via IFT-independent mechanisms. Our results reveal unexpected complexity in the contribution of IFT and cilia organization to the regulation of responses even within a single chemosensory neuron type and establish a critical role for these processes in the precise modulation of olfactory behaviors.

## Introduction

Individual sensory neurons are highly specialized to respond to defined environmental stimuli. Although subsets of polymodal sensory neurons have been described across organisms [1–4],

**Data Availability Statement:** Underlying data for behavioral and calcium imaging experiments are provided in https://doi.org/10.5281/zenodo.

13748735. All other data are within the paper and its Supporting Information files.

**Funding:** This work was supported in part by the National Institutes of Health (www.nih.gov) (R35 GM122463 – P.S. and F32 DC018453 – A.P.). The funders had no role in study design, data collection and analysis, decision to publish, or preparation of the manuscript.

**Competing interests:** The authors have declared that no competing interests exist.

**Abbreviations:** CTCF, corrected total cell fluorescence; EV, extracellular vesicle; FRAP, fluorescence recovery after photobleaching; IFT, intraflagellar transport; NGM, nematode growth medium; PCMC, periciliary membrane compartment.

the majority of sensory neuron types are unimodal. Photoreceptors respond only to light, and olfactory neurons detect and respond only to volatile odorants. However, even unimodal sensory neurons exhibit extensive functional heterogeneity. Cone photoreceptor subtypes respond to distinct wavelengths of light [5,6], and individual olfactory and gustatory neuron types sense and drive behavioral responses to unique chemical subsets [7–11]. This functional diversity is largely mediated via the cell type-specific expression of sensory receptors and other signal transduction molecules within each neuron type [9,12–14]. Whether other features also contribute to diversification of response properties within and across sensory neuron subtypes remains to be fully described.

A defining morphological feature of many sensory neurons is the localization of sensory signal transduction molecules to microtubule-based primary cilia. Although primary cilia are now known to be present on nearly all cell types in mammals including on central neurons [15–17], sensory neuron cilia are unique in that they exhibit remarkably varied and complex cell type-specific morphologies and are directly or indirectly exposed to the external environment [17,18]. Most cilia are built by the highly conserved process of intraflagellar transport (IFT), a motor-driven process that traffics ciliary structural and signaling proteins into and out of the cilia [19]. The structural complexity of photoreceptor ciliary outer segments allows for dense packing and organization of phototransduction molecules, thereby accounting in part for the remarkable ability of these neurons to efficiently capture photons [20,21]. In the vertebrate olfactory epithelium, olfactory receptors and transduction channels are localized to multiple cilia of varying numbers and lengths present at the distal ends of sensory neuron dendrites [22–24]. Although overall cilia length may be correlated with odorant sensitivity of these neurons [25], whether and how structural diversity in cilia architecture shapes the response profiles of individual olfactory neurons is unknown. Moreover, given the requirement of IFT for building cilia, it has been challenging to distinguish between the contributions of cilia morphology, and continuous IFT-mediated trafficking of signaling proteins, to sensory neuron response properties.

*Caenorhabditis elegans* sensory neurons provide an experimentally amenable system in which to correlate, and mechanistically describe, the role of cilia structure and IFT in regulating response properties. Only sensory neurons in *C. elegans* are ciliated, and as in other organisms, these cilia are built via IFT and house sensory transduction molecules [26,27]. Sensory cilia in *C. elegans* exhibit remarkably diverse morphologies which can be visualized and characterized on a neuron-by-neuron basis in single animals [27–29]. For instance, the 2 ASH nociceptive neurons in the bilateral amphid sense organs of the head each contain a single rod-like cilium, whereas the AWA olfactory neurons contain a highly branched cilium with a unique underlying cytoskeletal architecture [27,28]. Each sensory neuron pair responds to a range of stimuli within and across modalities, in part due to the expression of large numbers of sensory receptors in each neuron type [7,30,31]. Although loss of cilia in IFT mutant backgrounds is assumed to abolish all sensory responses in *C. elegans* [27,32–34], but see [35], the contribution of cilia and IFT to sensory transduction has not been analyzed systematically.

Here, we show that cilia architecture and IFT-mediated trafficking of signaling proteins differentially regulate response properties within and across 2 chemosensory neuron types in *C. elegans*. Via acute disruption of cilia structure but not IFT and vice versa, we show that cilia length but not continuous IFT is necessary for the responses to a subset of chemicals in the ASH nociceptive neurons. The roles of cilia shape and IFT are particularly complex in regulating odorant responses in the AWA olfactory neurons. We find that IFT is not required to maintain the complex AWA cilia architecture, but that prolonged IFT loss is associated with the formation of ectopic branches emanating from the periciliary membrane compartment (PCMC), regardless of the presence or absence of AWA cilia. Primary odorant responses are

maintained under these conditions in part due to the shunting of odorant receptors and a subset of signaling molecules into these PCMC branches. However, spatial segregation of olfactory receptors from a cilia-localized GPCR kinase results in odorant desensitization defects in the absence of IFT. We also find that while IFT-mediated removal of the diacetyl receptor from AWA cilia is necessary for adaptation to this odorant upon repeated exposure, adaptation to the odorant pyrazine is mediated via IFT-independent mechanisms. Our results reveal unexpected complexity in the role of cilia and IFT in shaping responses to ecologically important chemical cues both within and between 2 different chemosensory neuron subtypes, and indicate that these processes play a critical role in diversifying and regulating stimulus-specific neuronal and organismal responses.

## Results

### Mutations in IFT genes differentially affect chemosensory responses in the ASH nociceptive neurons

Eight of the 12 sensory neuron pairs including ASH in the amphid sense organs contain 1 or 2 rod-like cilia at their distal dendritic ends that are enclosed within a glial channel and are directly exposed to the environment [27–29] (Fig 1A). To test how truncation of cilia in IFT mutants affects neuronal responses, we examined chemical-evoked intracellular calcium dynamics in ASH neurons expressing GCaMP3. ASH responds to multiple aqueous and volatile nociceptive chemicals including glycerol and quinine, as well as high concentrations of isoamyl alcohol and 1-heptanol [36–38]. Mutations in the *osm-6*/IFT52 component of the IFT-B complex severely truncate ASH cilia (Fig 1A) [27]. ASH failed to respond to examined glycerol and quinine concentrations in *osm-6* mutants (Figs 1B, S1A and S1B). However, these neurons retained the ability to respond to higher concentrations of both isoamyl alcohol and 1-heptanol albeit with significantly decreased response amplitudes and altered response kinetics (Figs 1B, 1C, S1C and S1D). Adaptation of ASH to a subset of chemicals has previously been shown to be modulated by GABAergic signaling from surrounding glial cells [39,40]. We observed no changes in glycerol responses in ASH in either wild-type or *osm-6* mutants in the presence of exogenous GABA or the GABA$_A$ receptor antagonist bicuculline (S1E Fig), suggesting that mutations in IFT genes do not alter ASH responses via modulation of glial GABAergic signaling.

To determine whether there is a minimum cilium length above which responses to aqueous chemicals are maintained, we examined glycerol-evoked calcium dynamics in IFT mutants with different ASH cilia lengths. Mutations in the *daf-10*/IFT122 IFT-A complex protein also significantly truncate ASH cilia, whereas only the distal ciliary segments are lost in *osm-3* homodimeric kinesin-2 anterograde IFT motor mutants (Fig 1C) [27,41]. ASH cilia length is unaltered in animals mutant for the *kap-1* component of the heterotrimeric anterograde kinesin-II motor (Fig 1C). *kap-1* but not *daf-10* mutants retained the ability to robustly respond to glycerol, whereas responses in *osm-3* mutants were variable (Fig 1C). We infer that a minimum cilium length may contribute to robust responses to aqueous but not volatile chemicals in ASH. Alternatively, IFT may differentially regulate responses to these chemicals in ASH.

### Cilia length but not continuous IFT regulates glycerol responses in ASH

Mutations in IFT genes disrupt not only cilia structure but also IFT-mediated trafficking of signaling proteins, a subset of which is localized to the distal ciliary tip [42–46]. To decouple the contributions of IFT and cilia length to sensory signaling, we engineered a temperature-sensitive (*ts*) mutation in *osm-3*. The *fla8-2* mutation (F55S) in the motor domain of the highly

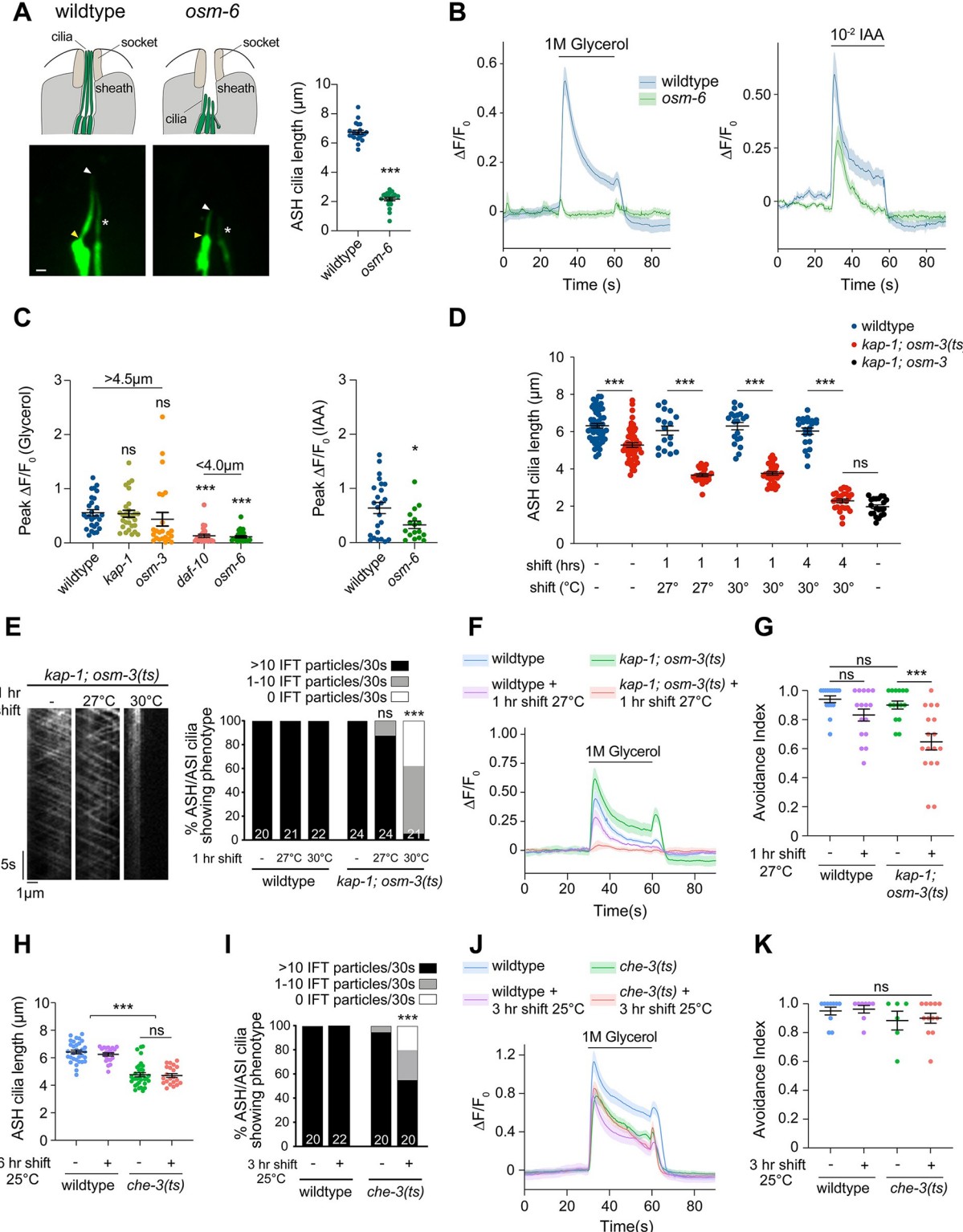

**Fig 1. Cilia length but not continuous IFT regulates glycerol responses in ASH.** **(A)** (Left) Cartoons (top) and representative images (bottom) of ASH cilia expressing *gfp* in wild-type and *osm-6(p811)* mutants under the *sra-6* promoter. Yellow/white arrowheads: cilia base/cilia tip; white asterisk: neighboring ASI cilium. Scale bar: 1 μm. (Right): Quantification of ASH cilia length in wild-type and *osm-6(p811)* adult hermaphrodites. ***: different from wild type at $P < 0.001$ (*t* test). **(B)** Mean GCaMP3 fluorescence changes in ASH to a 30 s pulse of the indicated odorants in wild-type and *osm-6(p811)* mutants. Shaded regions indicate SEM. $n \geq 17$ neurons each. For glycerol responses, data are

reported from the second stimulus pulse (see Methods); the 0 time point represents 90 s following initiation of imaging. (C) Quantification of peak fluorescence intensity changes in ASH expressing GCaMP3 to 1 M glycerol (left) or $10^{-2}$ dilution of IAA (right) in animals of the indicated genotypes. * and ***: different from wild type at $P < 0.05$ and $P < 0.001$, respectively (glycerol: one-way ANOVA with Tukey's multiple comparisons test; IAA: $t$ test). Numbers above indicate approximate ASH cilia lengths. (D) Quantification of ASH cilia lengths in the indicated conditions and genotypes. ***: different from corresponding wild type at $P < 0.001$ ($t$ test). (E) (Left) Representative kymographs of OSM-6:: split-GFP movement in ASH cilia for the indicated conditions in *kap-1(ok676 lf); osm-3(oy156ts)* mutants. (Right) Percentage of cilia exhibiting OSM-6::split-GFP movement in 30 s (see S1 Table). ***: different at $P < 0.001$ from *kap-1; osm-3(ts)* prior to temperature upshift (Fisher's exact test). (F, J) Mean GCaMP3 fluorescence changes in ASH to a 30 s pulse of 1 M glycerol for the indicated genotypes and conditions. The *che-3 (nx159ts)* allele was used in J. Shaded regions indicate SEM. $n \geq 15$ neurons each. For glycerol responses, the 0 time point represents 90 s following initiation of imaging (see Methods). (G, K) Quantification of avoidance to 8 M Glycerol for the indicated genotypes and conditions; 0 and 1 indicate 0% and 100% avoidance, respectively. ***: different at $P < 0.001$ between indicated values (one-way ANOVA with Tukey's multiple comparisons test). (H) Quantification of ASH cilia lengths in the indicated genotypes and conditions. ***: different at $P < 0.001$ between indicated values (one-way ANOVA with Tukey's multiple comparisons test). (I) Percentage of cilia exhibiting OSM-6::split-GFP movement in the indicated genotypes and conditions. ***: different at $P < 0.001$ from *che-3(ts)* prior to temperature upshift (Fisher's exact test). Each dot in the scatter plots is the measurement from a single ASH neuron except in G and K where each dot indicates a single assay of 10 animals each. Horizontal lines in all plots indicate the mean. Errors are SEM. Data shown are from a minimum of 2 to 3 independent experiments. ns: not significant. Underlying data are provided in https://doi.org/10.5281/zenodo.13748735. IAA, isoamyl alcohol; IFT, intraflagellar transport.

conserved FLA8 ciliary kinesin motor protein of *Chlamydomonas reinhardtii* results in rapid loss of flagella upon a shift from permissive to the restrictive temperature [47–49]. An *osm-3 (oy156)* mutant engineered to carry the corresponding F55S mutation (S2A Fig) was defective in its ability to uptake a lipophilic dye within 4 h of a shift from 20°C to 30°C similar to the dye uptake defects of *osm-3(p802)* loss-of-function (*lf*) mutants (S2B Fig). Since dye-filling defects are associated with structural alterations in a subset of amphid sense organ sensory cilia [27,50], these observations suggest that *osm-3(oy156)* may be a *ts* allele.

We next examined ASH cilia length and IFT in *osm-3(oy156)* mutants upon shifting to the restrictive temperature for different periods of time. Since OSM-3 acts redundantly with the heterotrimeric kinesin-II motor to build the middle segments of channel cilia [51], we examined ASH cilia in *kap-1(ok676 lf); osm-3(oy156)* double mutants. ASH cilia were significantly truncated within approximately 1 h of shift to either 27°C or 30°C in *osm-3(oy156)* mutants and were truncated to the same extent as in *kap-1(ok676); osm-3(p802) lf* double mutants after 4 h at 30°C (Fig 1D). To visualize movement of IFT proteins at endogenous levels in ASH cilia, we tagged the *osm-6* IFT-B gene with the split-GFP reporter $GFP_{11}$ (*oy166* allele) via gene editing. ASH cilia length and dye-filling was unaltered in these animals (S2C Fig) indicating that the tag did not disrupt *osm-6* function. We reconstituted GFP via expression of the $GFP_{1-10}$ fragment in ASH and monitored OSM-6 movement. OSM-6::split-GFP moved at speeds similar to those of overexpressed OSM-6::GFP reported previously (Figs 1E and S2D and S1 Table). Both anterograde and retrograde movement were markedly reduced upon a 1-h shift to 30°C, but continued, albeit at a reduced frequency after shifting to 27°C for 1 h (Fig 1E and S1 Table). These observations confirm that *osm-3(oy156)* represents a *bona fide ts* allele (henceforth referred to as *osm-3(ts)*).

Shifting these double mutants to the semi-permissive temperature of 27°C provides a time window within which to assess the effects of cilia length truncation on glycerol responses in ASH in the presence of continued IFT. While ASH responded robustly to glycerol in *kap-1(lf); osm-3(ts)* animals at 20°C, these responses were eliminated upon an acute shift to 27°C for 1 h (Fig 1F). Moreover, while these animals avoided a ring of glycerol at the permissive temperature similar to wild-type animals, they escaped the ring upon exposure to 27°C for 1 h (Fig 1G). To determine how the converse conditions in which cilia length is maintained but IFT is acutely blocked affect glycerol responses, we examined *che-3(nx159ts)* mutants [52]. *che-3* encodes the ciliary retrograde IFT dynein-2 motor, and the *ts* allele has previously been shown to disrupt both anterograde and retrograde IFT within 3 h of a temperature upshift

without significant effects on cilia length [52]. Consistently, while ASH cilia length was slightly but significantly shorter in *che-3(ts)* mutants at the permissive temperature of 15˚C, length was not further affected even after growth at 25˚C for 6 h (Fig 1H). However, movement of OSM-6::split-GFP was significantly reduced within 3 h following a temperature upshift (Fig 1I and S1 Table). Neither glycerol-evoked calcium responses nor behavioral avoidance of glycerol were affected in *che-3(ts)* mutants under these conditions (Fig 1J and 1K). Together, these results indicate that glycerol responses in ASH require a minimum cilium length but not continuous IFT.

## Mutations in IFT genes differentially affect responses to volatile odorants in the AWA olfactory neurons

We next examined the contributions of cilia morphology and IFT to odorant responses in the AWA olfactory neurons which contain highly elaborate arborized cilia whose distal ends are embedded within glial processes [27–29]. These neurons mediate responses to a panel of bacterial food-related attractive volatile odorants including diacetyl and pyrazine [53]. It has previously been reported that truncation of AWA cilia in IFT mutants does not result in loss of the primary response to diacetyl but instead leads to defects in response desensitization (defined here as the decay of the response) and adaptation (defined here as progressively decreasing responses upon repeated stimulation; previously referred to as habituation [35]). Responses to other AWA-sensed odorants in IFT mutants were not examined. We investigated the effects of IFT mutations on responses to diacetyl and pyrazine as well as structurally related odorants in AWA.

AWA cilia as visualized via expression of a myr-GFP membrane-associated reporter were severely truncated in both *osm-6(p811)* and *kap-1(ok676); osm-3(p802) lf* double mutants (Fig 2A). We also noted a large number of ectopic processes emanating from the PCMC at the base of the AWA cilia (henceforth referred to as PCMC branches) (Fig 2A); these are discussed further below. As reported previously [35], these mutants retained primary responses over a broad range of diacetyl concentrations but exhibit desensitization defects (Figs 2B and S3A). Similarly, *osm-6* mutants continued to respond robustly to 2-heptanone [54] at different concentrations but exhibited altered desensitization rates (S3B Fig). In contrast, response amplitudes in AWA to different concentrations of both pyrazine and 2-methylpyrazine [55] were markedly decreased in IFT mutants (Figs 2C, S3C and S3D).

To determine whether the observed response phenotypes result in defects in attraction to these odorants, we examined behavioral responses to diacetyl and pyrazine in microfluidics behavioral arenas [38,56] that allow assessment of attraction to the same chemical concentrations used in calcium imaging experiments. Wild-type animals were robustly attracted to both odorants and accumulated within the central odorant stripe over the 20 min assay period (Fig 2D and 2E). While *osm-6* mutants retained the ability to be attracted to a range of diacetyl concentrations, *kap-1(ok676); osm-3(p802)* double mutants retained partial attraction to high but not low diacetyl concentrations (Figs 2D, 2E and S3E) [35]. Consistent with reduced pyrazine response amplitudes, *osm-6(p811)* and *kap-1(ok676); osm-3(p802)* mutants were indifferent to all tested concentrations of pyrazine (Figs 2D, 2E and S3E). We conclude that truncation of cilia and/or loss of IFT differentially affects AWA responses to distinct volatile odorants.

## Differential expression of odorant receptors upon chronic loss of IFT may drive distinct olfactory responses in AWA

To begin to explore how mutations in IFT genes result in odorant-specific response phenotypes in AWA, we hypothesized that *lf* mutations in IFT genes may alter the localization and/

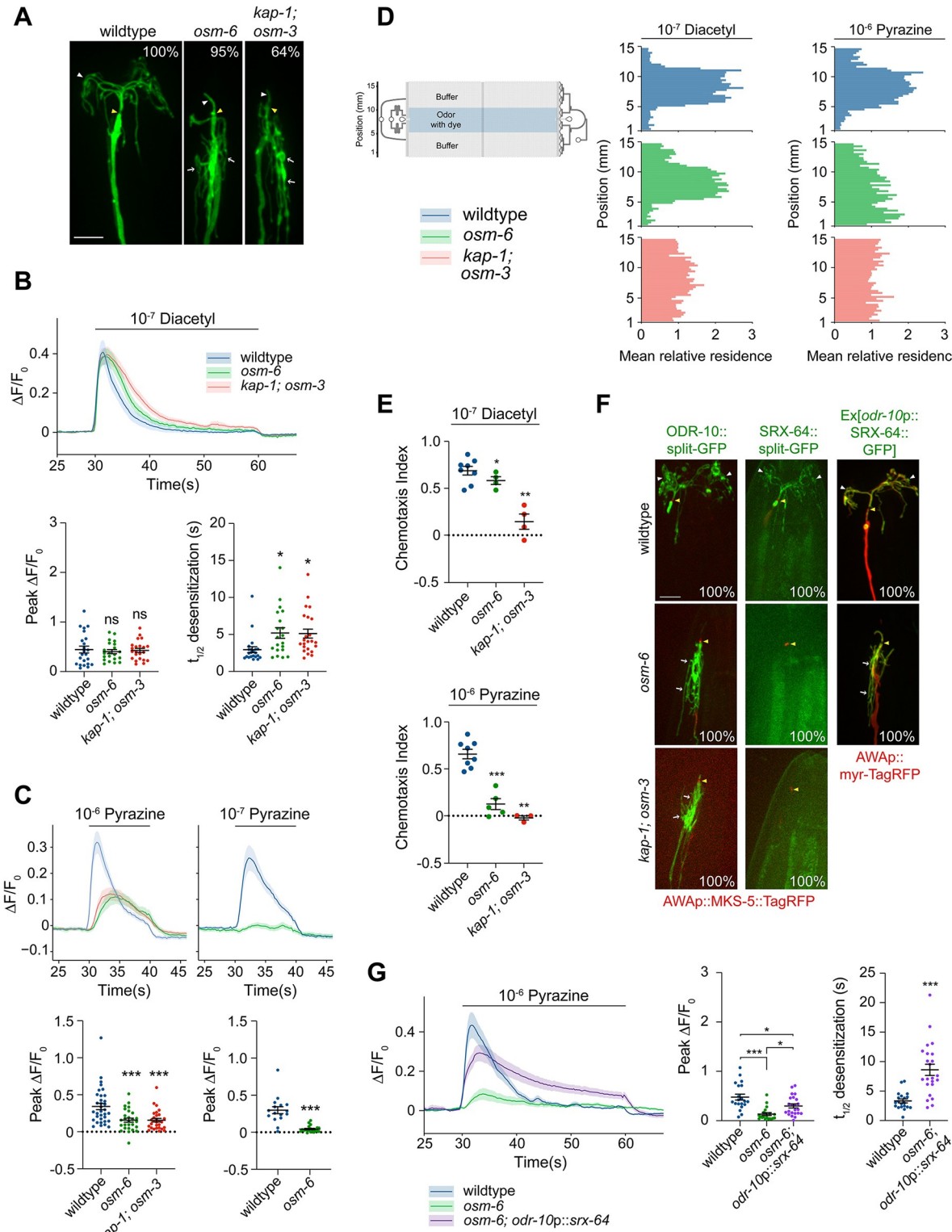

**Fig 2. Chronic disruption of IFT differentially affects the expression of odorant receptors in AWA to drive distinct odorant responses.** (**A**) Representative images of AWA cilia and PCMC branches (white arrows) in the indicated strains. AWA was visualized via expression of AWAp::*myr-GFP*. Numbers at top right indicate the percentage of neurons exhibiting the shown phenotype; $n \geq 20$. (**B**) (Top) Mean GCaMP fluorescence changes in AWA in response to a 30 s pulse of diacetyl. Shading indicates SEM. (Bottom) Quantification of peak fluorescence intensities (left) and desensitization rates (right) for the diacetyl response in AWA in animals of the indicated genotypes. >95%

of traces fit t$_{1/2}$. *: different from wild type at $P < 0.05$ (one-way ANOVA with Tukey's multiple comparisons test). **(C)** (Top) Mean GCaMP fluorescence changes in AWA in response to a 10 s pulse of pyrazine at the indicated concentrations. Shading indicates SEM. (Bottom) Quantification of peak fluorescence intensities for the pyrazine responses in AWA in animals of the indicated genotypes. ***: different from wild type at $P < 0.001$ (left: one-way ANOVA with Tukey's multiple comparisons test, right: $t$ test). **(D)** (Left) Schematic of a behavioral microfluidics device (from [38]). (Right) Average histograms showing mean relative $x$-$y$ residence of animals of the indicated genotypes in the behavioral device with a central stripe of the indicated odorants. **(E)** Chemotaxis indices calculated from the behavioral assays shown in D. *,**, ***: different from wild type at $P < 0.05$, $P < 0.01$, $P < 0.001$ ($t$ test; each mutant strain was assayed in parallel with wild-type controls in the same device in each assay). **(F)** Representative images of ODR-10::split-GFP, SRX-64::split-GFP, or *odr-10*p::SRX-64::GFP localization in AWA in the indicated strains expressing the shown reporters. Numbers at bottom right indicate the percentage of animals showing the phenotype. $n \geq 20$ for each. **(G)** (Left) Mean GCaMP fluorescence changes in AWA in response to a 30 s pulse of pyrazine. Shading indicates SEM. (Middle) Quantification of peak fluorescence intensities and (right) desensitization rates for the diacetyl response in AWA in animals of the indicated genotypes. 100% of traces fit t$_{1/2}$. * and ***: different from indicated genotype at $P < 0.05$ and $P < 0.001$ (middle: one-way ANOVA with Tukey's multiple comparisons test, right: $t$ test). In all images, yellow/white arrowheads indicate cilia base/cilia tip, arrows indicate PCMC branches. Scale bars: 5 μm. Each dot in the scatter plots in B, C, and G is the measurement from a single AWA neuron; each dot in the plots in E is the chemotaxis index from a single behavioral assay. Horizontal lines in all plots indicate the mean. Errors are SEM. Data shown are from a minimum of 2 to 3 independent experiments. ns: not significant. Alleles used were: *osm-6 (p811)*, *kap-1(ok676)*, and *osm-3(p302)*. All calcium imaging was initiated 30 s prior to stimulus onset; responses from 5 s prior to stimulus onset are shown. Underlying data are provided in https://doi.org/10.5281/zenodo.13748735. IFT, intraflagellar transport; PCMC, periciliary membrane compartment.

or levels of the *odr-10* and *srx-64* diacetyl and pyrazine receptors, respectively, in AWA [57,58]. Endogenously tagged and functional ODR-10 [59] and SRX-64 proteins (S4A Fig) localized specifically to the AWA cilia in wild-type animals (Fig 2F). We detected expression of ODR-10::split-GFP both in the truncated cilium and the PCMC branches in *osm-6* and *kap-1; osm-3* mutants (Fig 2F). However, levels of SRX-64::split-GFP were markedly down-regulated, and this fusion protein was barely detectable in either the ciliary stub or PCMC branches in IFT mutants (Figs 2F and S4B) likely accounting for the marked decrease in their pyrazine responses. This down-regulation appears to be mediated via transcriptional mechanisms since expression of an endogenous SRX-64::SL2::split-GFP transcriptional reporter was also significantly decreased in an IFT mutant (S4C Fig). Consistent with this notion, expression of *srx-64::gfp* under the *odr-10* promoter restored SRX-64 protein levels in *osm-6* mutants, with localization in both the ciliary stub and the PCMC branches (Fig 2F). Pyrazine responses were also restored in these animals, and as in the case of diacetyl, these responses exhibited desensitization defects (Fig 2G). These results indicate that reduced expression of SRX-64 likely accounts for the pyrazine response defect in IFT *lf* mutant backgrounds. These observations further imply that upon localization of both receptors to the cilia stub and the PCMC branches, *lf* mutations in IFT genes primarily affect odorant desensitization in AWA.

## Odorant desensitization defects upon loss of IFT correlate with the presence of ectopic PCMC branches regardless of cilia morphology

We next tested whether the odorant desensitization defects in IFT *lf* mutants arise due to loss of IFT, disruption of AWA cilia morphology, and/or the formation of PCMC branches. To do so, we acutely inhibited IFT in *kap-1; osm-3(ts)* mutants by performing temperature shift experiments for different time periods and quantified odorant response properties.

Movement of endogenously tagged OSM-6::split-GFP could be observed only within the primary stalk and very rarely within the thin branches of AWA cilia (S1 Movie) [60]. Although we observed reduced IFT in *kap-1; osm-3(ts)* double mutants in AWA even at the permissive temperature, all IFT was abolished within 1.5 h in these cilia following a temperature upshift to 30˚C (Fig 3A and S2 Table). However, AWA cilia structure remained grossly unaffected even upon growth of animals at 30˚C for up to 24 h (Figs 3B and S5A), suggesting that IFT is not necessary for maintenance of AWA cilia architecture in adults. Few PCMC branches were observed following a 1.5 h upshift (Fig 3B). At this time point, responses to both diacetyl and

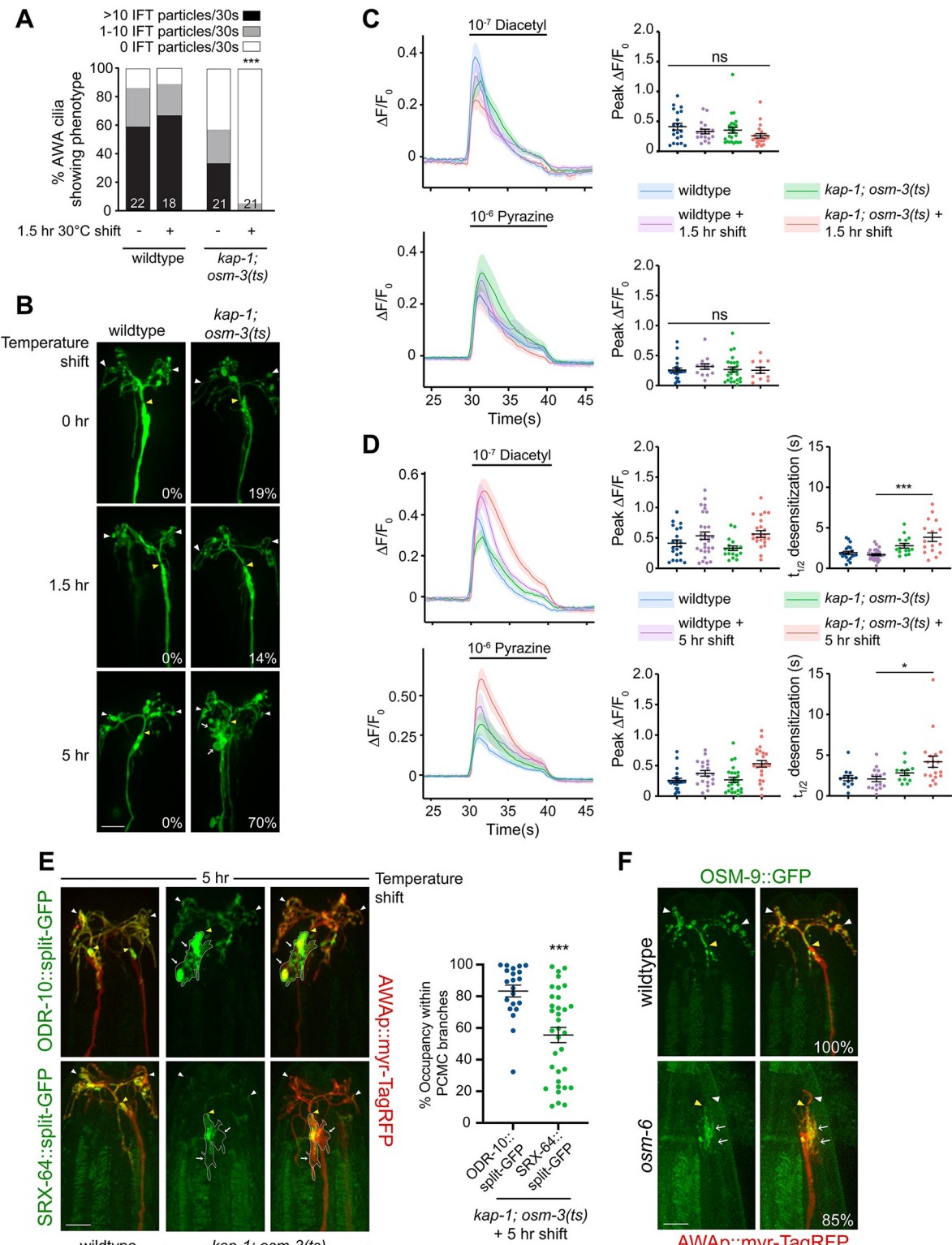

**Fig 3. The formation of ectopic PCMC branches in AWA may underlie odorant desensitization defects upon loss of IFT. (A)** Percentage of animals of the indicated genotypes exhibiting OSM-6::split-GFP movement within the proximal AWA cilia stalk under the shown temperature shift conditions. ***: different at $P < 0.001$ from *kap-1; osm-3(ts)* prior to temperature upshift (Fisher's exact test). **(B)** Representative images of AWA cilia expressing *gpa-4Δ6*p::*myr-GFP* in the indicated strains and temperature shift conditions. Numbers at bottom right indicate the percentage of AWA neurons exhibiting PCMC branches. $n \geq 20$. **(C, D)** Mean GCaMP fluorescence changes

(left), quantification of peak fluorescence intensities (C—right, D—middle) and desensitization rates (D—right) in AWA in response to a 10 s pulse of the indicated odorants in animals of the shown genotypes upon temperature shift for 1.5 h (C) or 5 h (D). At least 67% of traces fit $t_{1/2}$. Shading indicates SEM. * and ***: different at $P < 0.05$ and $P < 0.001$ from corresponding control (one-way ANOVA with Tukey's multiple comparisons test). Control data prior to temperature upshift are repeated in C and D. **(E)** (Left) Representative images of ODR-10::split-GFP (top) or SRX-64::split-GFP (bottom) in wild-type or *kap-1(ok676); osm-3(oy156ts)* mutants upon the indicated temperature shift conditions. (Right) Quantification of ODR-10::split-GFP or SRX-64::split-GFP occupancy within AWA PCMC branches after a 5-h temperature shift. ***: different at $P < 0.001$ from ODR-10::split-GFP occupancy (*t* test). **(F)** Representative images of OSM-9::GFP localization in the AWA sensory endings of wildtype and *osm-6(p811)* mutants. Numbers at bottom right indicate the percentage of neurons exhibiting the shown phenotype; $n = 20$. In all images, yellow/white arrowheads indicate cilia base/cilia tip, arrows indicate PCMC branches. Each dot in scatter plots is the measurement from a single AWA neuron. Horizontal lines indicate the mean. Errors are SEM. Data shown are from a minimum of 2 to 3 independent experiments. ns: not significant. Scale bars: 5 μm. All calcium imaging was initiated 30 s prior to stimulus onset; responses from 5 s prior to stimulus onset are shown. Underlying data are provided in https://doi.org/10.5281/zenodo.13748735. IFT, intraflagellar transport; PCMC, periciliary membrane compartment.

pyrazine were similar to those in wild-type animals (Fig 3C), suggesting that continuous IFT is not necessary for regulation of either the primary response or odorant desensitization in AWA.

Following a 5-h shift of *kap-1; osm-3(ts)* to 30°C, we noted multiple PCMC branches in AWA although cilia morphology remained unaltered (Fig 3B), indicating that prolonged loss of IFT is correlated with PCMC branch formation regardless of the presence or absence of cilia. Despite the prolonged loss of IFT, both diacetyl and pyrazine responses were maintained at this time point (Fig 3D). However, both odorant responses exhibited desensitization defects, with stronger defects observed for diacetyl (Fig 3D). We infer that odorant desensitization defects may arise due to the formation of PCMC branches upon prolonged loss of IFT, regardless of the presence or absence of AWA cilia.

Since pyrazine responses were maintained in *kap-1; osm-3(ts)* mutants even in the absence of IFT for >5 h, we examined the levels and localization of the endogenously tagged SRX-64 pyrazine receptor in AWA under these conditions. As expected, ODR-10::split-GFP was again present in both the AWA cilia and throughout the PCMC branches (Fig 3E). SRX-64::split-GFP levels in AWA cilia were reduced although the fusion protein was still detectable in cilia (Fig 3E). SRX-64::GFP was also mislocalized to the PCMC branches under these conditions (Fig 3E) although the distribution of SRX-64::split-GFP in the PCMC branches was more variable, with this protein occupying only restricted domains in a subset of these branches (Fig 3E).

We tested whether ciliary proteins in addition to odorant receptors are also shunted into these PCMC branches in the absence of IFT. An EBP-2::split-GFP reporter was present in the branches in *osm-6* mutants indicating that these branches contain microtubules that appear to be oriented plus-end out (S5B Fig and S2 Movie). While the branches in *osm-6* mutants did not contain the KAP-1 or OSM-3 kinesin motors, or the ciliary GTPase ARL-13 (S5C Fig), a subset of ciliary OSM-9 TRPV and ODR-3 Gα proteins required for diacetyl and pyrazine signal transduction [61–63] was also mislocalized to these branches (Figs 3F and S5D). These results indicate that multiple ciliary signal transduction proteins are mislocalized to PCMC branches in the absence of IFT in AWA.

## Spatial segregation of odorant receptors and the GRK-2 GPCR kinase may account for the odorant desensitization defects upon prolonged loss of IFT

We next investigated the mechanistic basis for the observed odorant desensitization defects upon formation of PCMC branches. Rapid response desensitization following ligand binding to GPCRs is in part mediated via phosphorylation of cytoplasmic residues by GRK GPCR kinases [64–66]. The GRK-2 kinase has previously been implicated in modulating sensory

behaviors in *C. elegans* [67–69]. We found that while *grk-2* mutants responded to both diacetyl and pyrazine albeit with reduced amplitudes, these mutants exhibited desensitization defects to both odorants (Figs 4A, 4B and S6A). The *grk-2* diacetyl response phenotype was fully rescued upon expression of a wild-type reporter-tagged *grk-2* cDNA specifically in AWA (Figs 4A and S6A). *grk-2* mutants also exhibited defects in their ability to be attracted to diacetyl, and to a lesser extent pyrazine, in microfluidics behavioral arenas (Fig 4C and 4D). Although AWA cilia exhibited morphological defects, we did not observe PCMC branches in *grk-2* mutants (S6B Fig). These observations suggest that GRK-2 may target both ODR-10 and SRX-64 to modulate diacetyl and pyrazine desensitization in AWA.

Localization of a functional GRK-2::tagRFP fusion protein expressed under an AWA-specific promoter was restricted to the AWA cilia (Fig 4E) similar to the localization pattern of ODR-10 and SRX-64. However, unlike the odorant receptors, GRK-2 remained in the cilia and did not mislocalize to the ectopic branches in either *osm-6(p811)* IFT mutants, or in *kap-1; osm-3(ts)* mutants at the restrictive temperature (Fig 4E). Levels of GRK-2::tagRFP in the cilium were unaltered in the kinesin double mutants at the restrictive temperature (Fig 4E). We infer that spatial segregation of GRK-2 in the cilia from ODR-10 and SRX-64 localized to the PCMC branches may underlie the odorant desensitization defects observed upon prolonged or acute block of IFT in AWA (Fig 4F). These defects may be stronger for the diacetyl response due to the higher levels of ODR-10 as compared to SRX-64 localized to the PCMC branches (Fig 3E). These observations suggest that correct spatial distribution and organization of signaling components in AWA cilia may shape odorant response dynamics.

## Adaptation to diacetyl and pyrazine is mediated via distinct ciliary mechanisms in AWA

In addition to affecting desensitization, previous work showed that chronic loss of IFT also affects diacetyl response adaptation [35], although pyrazine responses were not examined. We next tested the contribution of cilia and ciliary trafficking mechanisms to odorant adaptation in AWA.

Responses to diacetyl showed robust adaptation upon exposure to repeated pulses of odorant (Figs 5A and S7A) [35]. Consistent with previous observations examining IFT mutants [35], *kap-1; osm-3* mutants showed strong diacetyl response adaptation defects (S7A Fig). To determine the mechanistic basis of this defect, we measured adaptation in *kap-1; osm-3(ts)* mutants prior to and following temperature shift. While diacetyl adaptation in *kap-1; osm-3 (ts)* mutants was similar to that of wild-type animals at the permissive temperature (Fig 5A), these animals exhibited adaptation phenotypes similar to those of *kap-1; osm-3 lf* mutants upon a shift to 30°C for 1.5 h (Fig 5A). Since only IFT is blocked with no obvious effects on either cilia or PCMC branches at this time point (see Fig 3A and 3B), we infer that continuous IFT is important for diacetyl adaptation. In contrast, while responses to pyrazine also showed robust adaptation upon repeated pulses of odorant, acute disruption of IFT had no effect on pyrazine adaptation (Fig 5A) indicating that adaptation to pyrazine is mediated via a distinct mechanism.

To identify the pathways by which IFT mediates diacetyl adaptation, we first attempted to examine whether ODR-10 transport is IFT-dependent in AWA cilia. Single molecule imaging has shown that GPCRs exhibit multiple modes of movement within cilia including diffusion and motor-driven directional transport, making it challenging to observe GPCR trafficking using ensemble measurements [70,71]. To determine whether ODR-10 is mobile within AWA cilia, and if this mobility is in part IFT dependent, we instead photobleached the primary ciliary stalk together with the transition zone of AWA cilia expressing ODR-10::split-GFP in

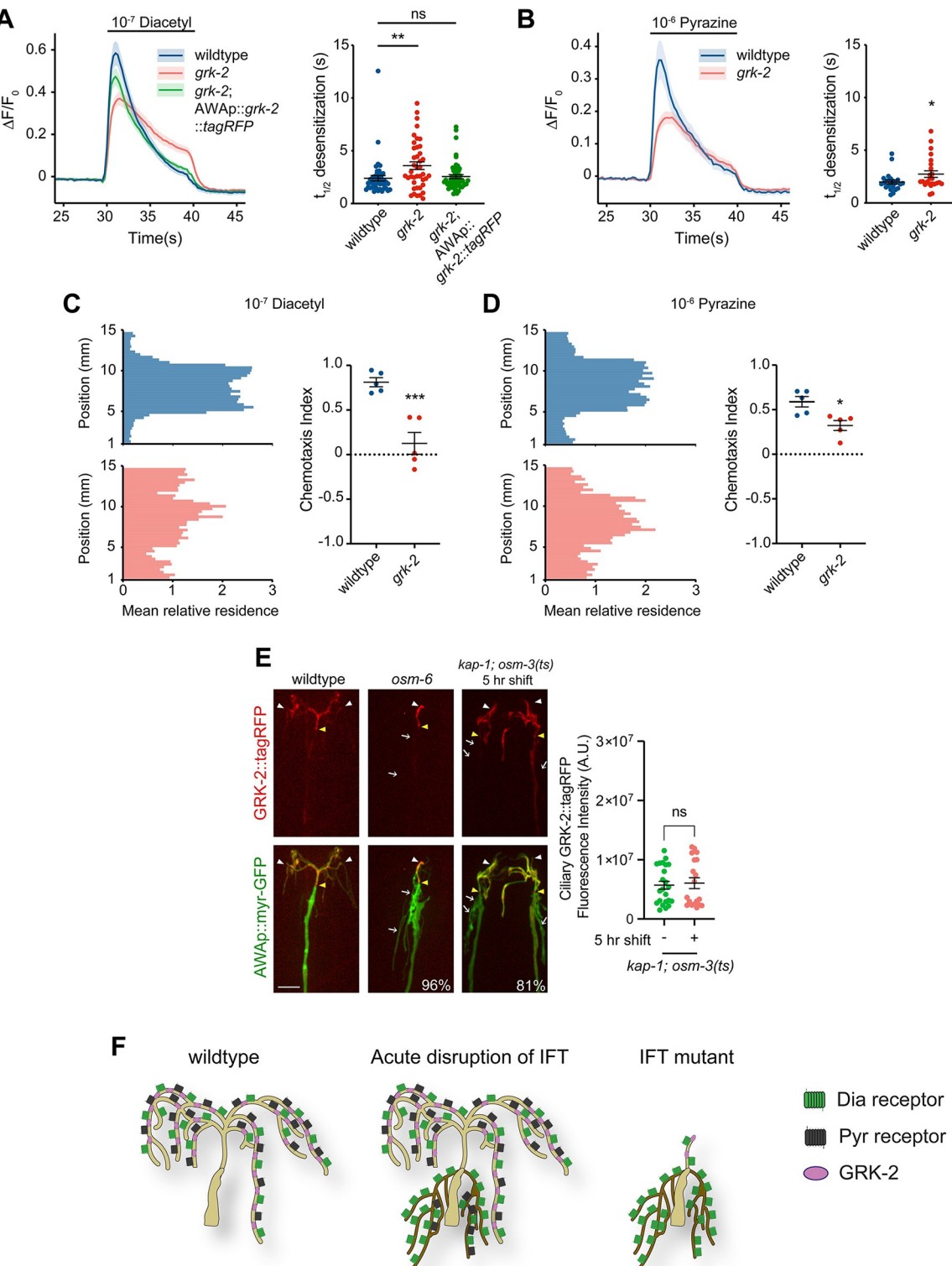

**Fig 4. Altered distribution of odorant receptors and the GRK-2 GPCR kinase may underlie response desensitization defects in the prolonged absence of IFT. (A, B)** (Left) Mean GCaMP fluorescence changes in AWA in response to a 10 s pulse of diacetyl (A) or pyrazine (B). Shading indicates SEM. The *grk-2(gk268)* allele was used. Wild-type *grk-2* sequences were expressed in AWA under the *gpa-4Δ6* promoter (A). (Right) Odorant desensitization rates in response to diacetyl (A) or pyrazine (B). At least 80% of traces fit $t_{1/2}$. * and **: different from wild type at $P < 0.05$ and $P < 0.01$ (one-way ANOVA with Tukey's multiple comparisons test in A; *t* test in B).

All calcium imaging was initiated 30 s prior to stimulus onset; responses from 5 s prior to stimulus onset are shown. **(C, D)** (Left) Average histograms showing mean relative *x-y* residence of animals of the indicated genotypes in microfluidics behavioral assays containing a central odorant stripe. (Right) Chemotaxis indices calculated from behavioral assays shown in C and D. Each dot is the chemotaxis index from a single assay. Horizontal lines indicate mean. Errors are SEM. * and ***: different at $P < 0.05$ and $P < 0.001$ from wild type (*t* test). **(E)** Representative images (left) and quantification (right) of GRK-2::tagRFP in AWA cilia for the indicated genotypes and conditions. Numbers at bottom right indicate percentage of animals exhibiting GRK-2::tagRFP localization only in AWA cilia; $n \geq 21$. ns: not significant (*t* test). Each dot in the scatter plot is the value from a single AWA neuron. Horizontal lines indicate the mean. Errors are SEM. Scale bar: 5 μm. **(F)** Summary of the distribution and localization patterns of ODR-10, SRX-64, and GRK-2 within cilia and/or PCMC branches in wild-type conditions, with acute disruption of IFT, or in chronic *lf* IFT mutants. In wild type, all molecules are localized to cilia. Upon prolonged disruption of IFT in the adult, cilia morphology is unaffected but the sensory endings contain PCMC branches. Under these conditions, GRK-2 remains restricted to cilia, whereas both ODR-10 and SRX-64 mislocalize to PCMC branches, with higher levels of ODR-10 present in these branches. In chronic IFT *lf* mutants, cilia are severely truncated and PCMC branches are formed. GRK-2 remains cilia-localized, whereas ODR-10 is again mislocalized to PCMC branches. *srx-64* is downregulated in IFT *lf* mutants. Underlying data are provided in https://doi.org/10.5281/zenodo.13748735. IFT, intraflagellar transport; PCMC, periciliary membrane compartment.

wild-type and *osm-6* mutants. While wild-type animals showed partial recovery (34.9 ± 3.3%) of ciliary fluorescence within 40 s of photobleaching, little fluorescence recovery (14.0 ± 3.1%) was observed in *osm-6* in the cilia (Fig 5B), although the protein showed increased recovery in the PCMC branches (28.6 ± 3.6%) (S7B Fig). We infer that ODR-10 may be partly transported via IFT in the AWA cilia.

Response adaptation is mediated in part via removal of activated GPCRs from the plasma membrane via endocytosis followed by degradation or recycling [72–76]. In cilia, the ciliary BBSome complex acts as an adaptor to transport activated GPCRs marked by phosphorylation by GRKs or ubiquitination out of the cilium via retrograde IFT [77,78]. We observed increased ciliary ODR-10 levels in animals mutant for the *grk-2* GRK protein, and these levels were rescued upon AWA-specific expression of wild-type *grk-2* sequences (S7C Fig). Ciliary ODR-10 levels were similarly increased in *bbs-7* mutants (Fig 5C) [79]. We also noted a conserved BBSome interaction motif in Helix 8 of ODR-10 (S7D Fig) [80], although mutation of this motif resulted in failure of this protein to be trafficked to cilia (S7E Fig), likely due to overlap with sequences required for AP-1 clathrin adaptor-mediated anterograde dendritic transport [81]. Consistent with a role of these proteins in removal of activated ODR-10 out of cilia to mediate adaptation, both *grk-2(gk268)* and *bbs-7* mutants exhibited significant diacetyl adaptation defects (Figs 5D and S7F). The diacetyl adaptation phenotype of *grk-2 bbs-7* double mutants was similar to that of *grk-2* mutants alone (S7F Fig), implying that GRK-2 acts upstream of BBS-7 in this process. In contrast, the SRX-64 pyrazine receptor does not contain a canonical BBSome binding motif (S7D Fig), and ciliary levels of this protein were unaltered in *bbs-7* mutants (Fig 5C). Moreover, pyrazine adaptation was unaffected in *bbs-7* mutants (Fig 5D). Together, these observations support the notion that phosphorylated ODR-10 may be removed from AWA cilia via BBSome-mediated trafficking to mediate diacetyl adaptation, whereas activated SRX-64 is removed via BBSome- and IFT-independent mechanisms.

An alternative mechanism of GPCR removal from cilia is via ectocytosis from the cilia tip [82,83]. In *C. elegans*, extracellular vesicle (EV) production is enhanced upon overexpression of ciliary proteins or in specific signaling contexts, further suggesting that it may represent an alternative mode of ciliary protein removal [84,85]. We tested whether activated SRX-64 may be removed from AWA cilia via EVs to mediate pyrazine adaptation.

*C. elegans* ciliary EVs are constitutively released at a low level from both the ciliary tip into the environment and from the PCMC; PCMC-released EVs are phagocytosed by surrounding amphid sheath (AMsh) glia and accumulate in their cytoplasm [84,85]. Since the distal ends of AWA cilia are embedded within the AMsh glia, we reasoned that the majority of EVs would be phagocytosed by glia. In wild-type animals, we detected very few endogenously tagged

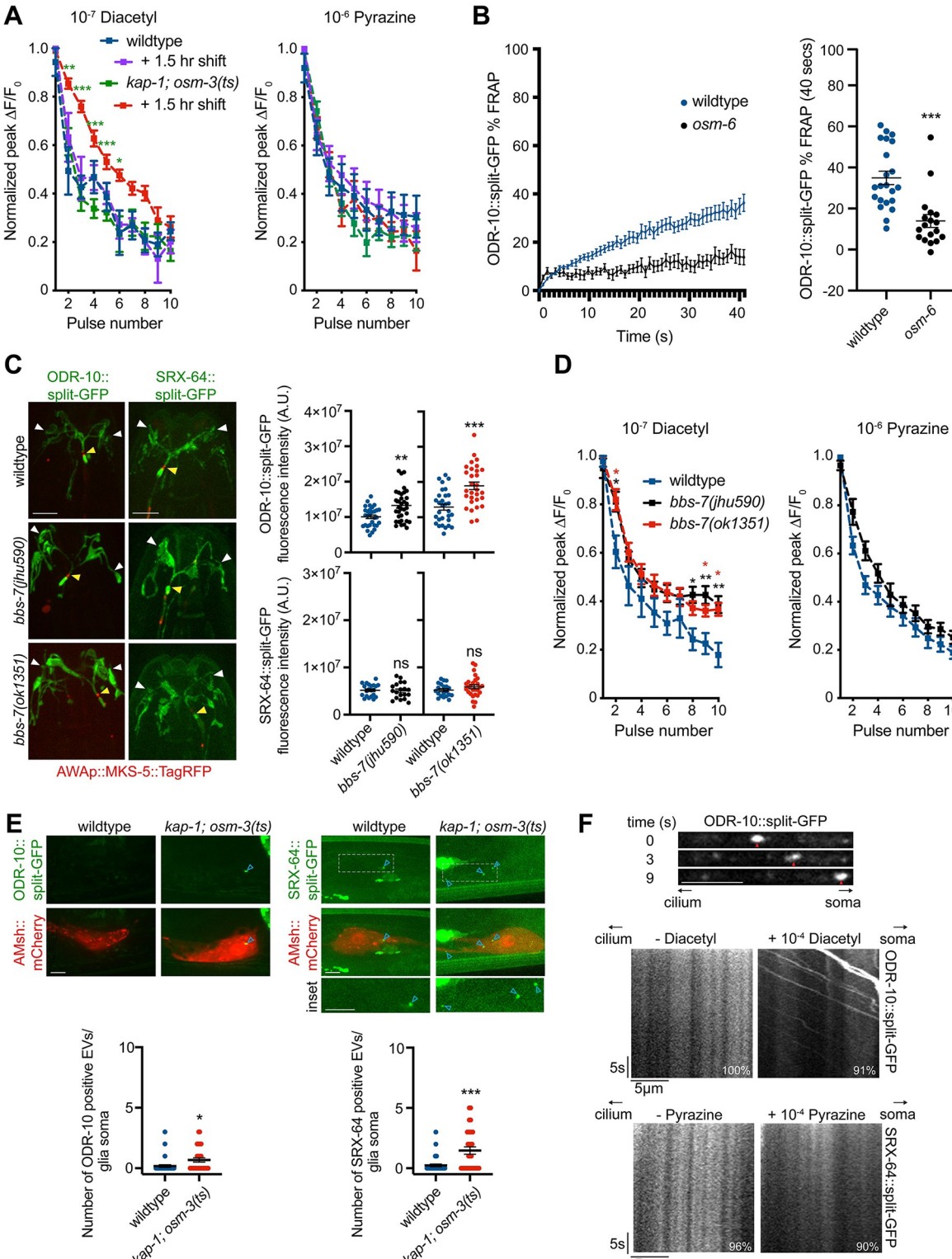

**Fig 5. Adaptation to diacetyl and pyrazine is mediated via IFT-dependent and IFT-independent mechanisms, respectively. (A)** Adaptation of peak AWA fluorescence upon repeated 10 s pulses of diacetyl (left) or pyrazine (right) in the indicated genotypes and conditions, normalized to the maximum peak $\Delta F/F_0$; $n \geq 11$ animals for each data point. *, ** and ***: different at $P < 0.05$, $P < 0.01$, and $P < 0.001$ from *kap-1; osm-3(ts)* prior to temperature upshift (second-way ANOVA with repeated measures). **(B)** (Left) Mean fluorescence recovery following photobleaching of the AWA ciliary stalk in wild-type and *osm-6(p811)* mutants expressing ODR-10::

split-GFP. (Right) Percentage recovery 40 s following photobleaching. ***: indicates different at $P < 0.001$ from wild type ($t$ test). **(C)** Representative images (left) and quantification (right) of ODR-10::split-GFP or SRX-64::split-GFP levels in wild-type and *bbs-7* mutants. ** and ***: different at $P < 0.01$ and $P < 0.001$ from corresponding wild type ($t$ test). **(D)** Adaptation of peak AWA fluorescence upon repeated 10 s pulses of diacetyl (left) or pyrazine (right) in the indicated genotypes, normalized to the maximum peak $\Delta F/F_0$; $n \geq 13$ animals for each data point. * and **: different at $P < 0.05$ and $P < 0.01$ from wild type (second-way ANOVA with repeated measures). **(E)** Representative images (top with insets shown below) and quantification (bottom) of ODR-10::split-GFP- or SRX-64::split-GFP-containing presumptive EVs (blue arrowheads) within the glia soma marked with *F16F9.3p::mCherry* in the indicated genotypes. * and ***: different at $P < 0.05$ and $P < 0.001$, respectively, from wild type ($t$ test). **(F)** (Top) Representative images of ODR-10::split-GFP fluorescence within the AWA dendrite upon addition of $10^{-4}$ diacetyl. Red arrow indicates location of a ODR-10::split-GFP punctum at the indicated time points following the start of recording. (Bottom) Representative kymographs of ODR-10::split-GFP or SRX-64::split-GFP movement in the presence or absence of added diacetyl or pyrazine, respectively. Number at bottom right indicates the percentage of animals exhibiting the shown movement; $n \geq 20$. In all images, yellow/white arrowheads indicate cilia base/cilia tip. Each dot in scatter plots is the measurement from a single AWA neuron. Horizontal lines indicate the mean. Errors are SEM. Data shown are from a minimum of 2 to 3 independent experiments. ns: not significant. Scale bars: 5 μm unless indicated otherwise. Underlying data are provided in https://doi.org/10.5281/zenodo.13748735. EV, extracellular vesicle; IFT, intraflagellar transport.

ODR-10::split-GFP or SRX-64::split-GFP puncta in the AMsh soma (Fig 5E). EV release from the PCMC is enhanced in the absence of IFT [84,85]. Although we could not examine SRX-64::split-GFP puncta in IFT *lf* mutants due to the down-regulation of SRX-64, we found that the number of SRX-64::split-GFP puncta increased in *kap-1; osm-3(ts)* mutants even at the permissive temperature presumably due to loss of kinesin-II (Fig 5E). We hypothesize that SRX-64 may be removed from AWA cilia via EVs to decrease pyrazine responses during odorant adaptation.

Overexpressed ODR-10 has been shown to traffic robustly in neuronal dendrites [81]. We reasoned that activated ODR-10 removed from cilia would be retrogradely trafficked from the cilia base to the AWA soma, and that this trafficking may be increased in the presence of added diacetyl. If SRX-64 is primarily ectocytosed from cilia, such trafficking would not be observed for this receptor. As shown in Fig 5F, we noted robust retrograde trafficking of ODR-10::split-GFP but not SRX-64::split-GFP from the cilia base to the soma upon addition of diacetyl or pyrazine, respectively (S3 and S4 Movies). Taken together, these results are consistent with the notion that ODR-10 and SRX-64 are regulated via distinct ciliary mechanisms to mediate response adaptation to their respective odor ligands (S7G Fig).

## Discussion

By acutely inhibiting IFT without disrupting cilia structure and vice versa, here we decouple the contributions of IFT and cilia morphology to the responses of 2 distinct chemosensory neuron types to diverse chemical cues. We find that cilia length but not continuous IFT is necessary for responses to a subset of chemicals sensed by ASH, whereas cilia and IFT are largely dispensable for primary odorant responses in AWA. Instead, IFT regulates the correct distribution and trafficking of signal transduction molecules at the sensory endings to differentially shape response dynamics to a subset of AWA-sensed odorants. Our results suggest that the diversity of ciliary mechanisms employed by individual neuron types may allow animals to precisely modulate their responses to specific chemical stimuli.

### The role of IFT in regulating chemosensory responses is distinct in ASH versus AWA cilia

The contribution of IFT to the regulation of odorant responses is unexpectedly complex across cilia types examined in this study. While IFT is necessary for the formation of both ASH and AWA cilia, anterograde IFT does not appear to be necessary for maintaining AWA cilia in adults. Acute disruption of IFT also does not truncate the flagella of the protozoan parasite *Trypanosoma brucei* [86] or chordotonal neuron cilia in *Drosophila* [87]. In ASH, continuous

IFT is dispensable for glycerol responses. In other *C. elegans* ciliated neurons with exposed rod-like cilia, signaling molecules are transported in part via IFT to the ciliary tip [42,43,88]. Once localized, these proteins appear to be remarkably stable and exhibit limited turnover or diffusion [42,89]; this stability may account for the maintenance of glycerol responses without continuous IFT. However, IFT plays a critical role in regulating responses to diverse ligands including the Hedgehog morphogen across species, suggesting that the contribution of IFT may be distinct for different stimuli and cell types [43,77,87,90–92]. Identification of the glycerol receptor and characterization of its localization upon acute and chronic inhibition of IFT will be necessary to address the underlying mechanisms in ASH.

As in ASH, continuous IFT is also dispensable for primary odorant responses in AWA. However, IFT shapes AWA response dynamics in 2 critical ways. First, continuous antero-grade IFT prevents the formation of PCMC branches. Branches from PCMCs have also been reported in mutants that are defective in glial phagocytosis of ciliary EVs resulting in buildup of ciliary proteins at the PCMC [85]. Prolonged block of IFT in AWA may result in the accumulation of large volumes of cilia-destined membrane and proteins at the PCMC [84,85,93,94]. The presence of receptors and other signaling molecules in these branches in the absence of cilia may account in part for the maintenance of the primary odorant response in IFT *lf* mutants. AWA-sensed odors such as diacetyl and pyrazine are attractive volatile odor-ants associated with bacterial food [95–98]. Although we were unable to selectively disrupt the formation of these PCMC branches, we suggest that the localization of odorant receptors and a subset of sensory signal transduction molecules in PCMC branches is a feature that allows ani-mals to retain the ability to find food even in the absence of cilia. The formation of these PCMC branches upon loss of IFT appears to be largely a feature of AWA but not of ASH or other neurons with simple cilia structures [27], possibly reflecting the larger volume of IFT cargo in complex cilia.

Second, IFT is critical for response adaptation upon repeated exposure to diacetyl in part via removal of ODR-10 from the cilia. The role of IFT-mediated retrograde trafficking of sig-naling proteins out of the cilium has now been extensively described both in vertebrate cilia and *Chlamydomonas* flagella [77,78]. Our results indicate that this retrograde trafficking plays a key role in odorant response adaptation. In contrast, adaptation to repeated presentation of pyrazine may be mediated by IFT-independent mechanisms, possibly via ectocytosis and EV production. Differential modes of activated receptor retrieval or loss from cilia have also been reported for neuropeptide receptors such as SSTR3 and NPY2R [82]. While it is currently unclear why different ciliary receptors are removed via different pathways, we speculate that the shuttling of different receptors to distinct trafficking pathways allows for more precise reg-ulation of specific odorant response dynamics.

## Cilia structure regulates chemosensory responses in a neuron- and odorant-specific manner

Upon severe cilia truncation as in IFT *lf* mutants, both ASH and AWA fail to respond to a sub-set of sensed chemicals while retaining responses to other odorants. A minimum cilium length regardless of the presence or absence of continuous IFT is necessary for responses to aqueous but not volatile odorants in ASH. Exposure of the distal endings of ASH and other glial chan-nel-localized neurons to the environment (see Fig 1A) is likely necessary to allow aqueous chemicals to directly interact with receptors and signaling molecules localized to the distal cili-ary tips [42,89]. In contrast, volatile odorant receptors are typically distributed throughout the sensory neuron cilia in *C. elegans* (e.g., [57,58,99]). Thus, high concentrations of volatile odor-ants such as isoamyl alcohol and 1-heptanol can likely diffuse through the channel to trigger a

response even in severely truncated ASH cilia, albeit with reduced efficiency. Neuronal cilia in the vertebrate brain have recently been shown to be integral components of synapses [100,101]. Cilia length and compartmentalized localization of neuropeptide and neurotransmitter receptors in cilia [102–104] may contribute to differential access by their cognate ligands in different cellular contexts.

In contrast, truncation of AWA cilia in IFT *lf* mutants decreased pyrazine but not diacetyl primary responses in AWA in part due to reduced SRX-64 but not ODR-10 expression. Increasing SRX-64 expression restores pyrazine responses in these mutants, indicating that as for diacetyl, neither cilia nor IFT are necessary for the primary odorant response in AWA. What then is the contribution of the complex cilia architecture to odorant responses in AWA? The organization and concentration of pathway components into signaling complexes is a key feature underlying the sensitivity and robustness, as well as the selectivity, of many signal transduction pathways including those within cilia [105–108]. Although we have not been able to manipulate AWA cilia structure in adults, we propose that the regulated localization and distribution of signaling components in the complex AWA ciliary branches is essential to ensure correct odorant response dynamics. In support of this notion, while GRK-2 levels and ciliary localization are unaltered upon prolonged conditional inhibition of IFT, these animals exhibit significant odorant desensitization defects, likely due to the spatial segregation of the relevant odorant receptors and GRK-2 in the cilia and the PCMC branches. Coordination of the spatial density and distribution of odorant receptors and regulatory molecules such as GRK-2 with the complex AWA cilia geometry may be essential for fine-tuning responses to drive robust and sensitive detection of critical food-related odorants.

### Concluding remarks

Our results together with previously published work [35] suggest that the role of cilia structure and IFT in regulating olfactory neuron responses in *C. elegans* is remarkably complex and distinct in 2 examined chemosensory neuron types. These differences can be attributed in part to the specific requirement of individual neuron types to detect chemicals over different concentration ranges. While neurons with simple rod-like cilia such as ASE respond to salts over only a 4-fold concentration range, AWA detects diacetyl over a $10^5$-fold range. The ability to detect odorants sensitively and robustly over this wide concentration range requires several specialized features of sensory signal transduction including rapid response desensitization and robust adaptation [35,109–113]. Results described here establish roles for cilia and sensory ending structure and IFT in regulating the remarkable response properties of AWA. However, our results also suggest that even within a single neuron type, responses are modulated in an odorant-specific manner, possibly reflecting the ethological hierarchy of different chemical cues. Some vertebrate and insect chemosensory neurons also express multiple receptors and respond to a range of chemicals (e.g., [9,114–117]). It will be of interest to establish whether similar diversity and complexity of ciliary mechanisms contribute to differentially shaping response properties in other chemosensory neuron types in *C. elegans* as well as across organisms.

### Materials and methods

#### *C. elegans* genetics

*C. elegans* strains were maintained on nematode growth medium (NGM) seeded with *Escherichia coli* OP50. Standard genetic techniques were used to construct all strains. The presence of mutations was confirmed using PCR-based amplification and/or sequencing. Experimental plasmid(s) were injected together with the *unc-122*p::GFP or *unc-122*p::dsRed coinjection

markers at 50 ng/μl or 60 ng/μl, respectively, to generate transgenic strains. Animals from at least 2 independent transgenic lines were examined from each injection, and 1 representative line was typically used for subsequent analyses. The same extrachromosomal array was examined in wild-type and mutant backgrounds for comparison. A complete list of strains used in this work is provided in S1 File. All experiments were performed with 1-day-old adult hermaphrodites.

## CRISPR/Cas9-mediated genome editing

Gene editing was performed using crRNAs, tracrRNAs, and Cas9 protein obtained from Integrated DNA technologies (IDT). $gfp_{11}$ sequences were inserted immediately before the stop codon. All insertions and point mutations were confirmed by sequencing.

*Generation of $gfp_{11}$ and SL2::$gfp_{11}$ knock-ins*: To generate the $gfp_{11}$ insertion alleles *osm-6 (oy166)*, *srx-64(oy193)*, and *srx-64(oy195)*, donor plasmids were first created with approximately 1 kb homology arms from genomic DNA flanking the $gfp_{11}$ or SL2::$gfp_{11}$ sequences (see S2 File for a list of plasmids used in this work). Donor templates including the approximately 1 kb homology arms were injected (100 ng/μl) together with crRNA (15 ng/μl), tracrRNA (20 ng/μl), and Cas9 protein (250 ng/μl) with *unc-122*p::*gfp* (50 ng/μl) or *dpy-10* crRNA and repair template (100 ng/μl and 15 ng/μl, respectively) as the co-injection marker. F1 progeny expressing the co-injection marker were isolated, and F2 animals were screened by PCR for the insertion and confirmed by sequencing.

*osm-6(oy166)* crRNA: 5′–TGATTTGTGATTGATTTGTC– 3′ (PSCR33)
*srx-64(oy193)* crRNA: 5′–AAAGCACATGCTTAAGCTCT– 3′ (PSCR26)
*srx-64(oy195)* crRNA: 5′–ATGTGATCACGTTTTTCTAC– 3′ (PSCR40)

To generate *ebp-2(oy178)*, a donor PCR product containing the desired $gfp_{11}$ insertion was amplified with 40 to 53 bp homology arms and injected together with crRNA (15 ng/μl), tracrRNA (20 ng/μl), and Cas9 protein (250 ng/μl) with *unc-122*p::*gfp* (50 ng/μl) as the co-injection marker. F1 animals expressing the co-injection marker were isolated, and F2 progeny were screened by PCR for the insertion and confirmed by sequencing.

*ebp-2(oy178)* crRNA: 5′–ACGGGGATAGGATAAGCAAT– 3′ (PSCR65)

*Generation of point mutations*: To generate *osm-3(oy156ts)* and *odr-10(oy194)*, donor oligo-nucleotides (IDT) containing the desired mutations along with 32 bp homology arms were injected together with crRNA (100 ng/μl), tracrRNA (167 ng/μl), and Cas9 protein (250 ng/μl). *dpy-10* crRNA (100 ng/μl) and repair template (15 ng/μl) were included as a co-injection marker, and F1 animals with Dpy or Rol phenotypes were isolated and their progeny screened by sequencing for the desired mutation. Dpy or Rol phenotypes were removed via outcrossing.

*osm-3(oy156ts)* crRNA: 5′–TCCATCGAACGTGAAGTCCT– 3′ (PSCR2)
donor oligonucleotide: 5′–CAGGTCAACCTGAACGCGCCGGACGGGGCGGCAAAG-GACTTCACGTCCGATGGAGCCTACTTTATGGATTCGACCGGCG– 3′
*odr-10(oy194)* crRNA: 5′–TCTTACTGAATATTGTTCTT– 3′ (PSCR24)
donor oligonucleotide: 5′–ATCCACTCATTCTGATTTTGATCATTCGTGATGCCGCA AGAACAATATTCAGTAAGAAGAAAACATATA– 3′

## Molecular biology

cDNAs (*odr-3*, *grk-2*, *osm-9*) were amplified from a pooled cDNA library obtained from wild-type animals [118] and cloned into plasmids containing the *gpa4Δ6* promoter [60] and fluorescent reporter sequences. Plasmids containing *sra-6*p::*myr-TagRFP*, *sra-6*p::$gfp_{1-10}$, *odr-10*p::*srx-64*::*gfp*, and *odr-10*p::*srx-64* were generated using traditional cloning methods. Split-GFP constructs were obtained from J. Kaplan (KP#3315). The *srx-64*::*gfp* sequence was obtained

from C. Chen [58]. All constructs were verified by sequencing. A full list of plasmids is provided in S2 File.

## Dye-filling

Stock solutions (1 mg/ml) of 1,1′-dioctadecyl-3,3,3′,3′-tetramethylindocarbocyanine perchlorate (DiI) in $N,N$-dimethylformamide (Sigma-Aldrich 468495) were stored at −20˚C. For assessing dye-uptake during temperature shift, animals were shifted to 30˚C for 3 h. Animals were soaked in 0.1 mg/ml dye at the designated temperature for an additional hour prior to analysis.

## Chemotaxis behavioral assays

Plate chemotaxis assays were performed as described previously [53] on 10-cm round plates. Briefly, 1 μl of the odorant diluted in ethanol or 1 μl ethanol as the control were placed along with 1 μl of 1 M sodium azide at each end of the assay plate. Animals were washed from large growth plates using S-basal, then washed twice with S-basal and once with Milli-Q water. Washed animals were transferred to chemotaxis plates and the number of animals at the odorant versus the ethanol control were counted after 1 h. Behaviors were examined in control and experimental animals in parallel for each day of analysis, and data reported are from 2 to 3 biologically independent assays performed over multiple days with at least 2 technical replicates per day.

## Osmotic avoidance assays

Osmotic avoidance assays were performed as described previously [33]. Briefly, 10 young adult hermaphrodites were transferred to an NGM plate without food for 5 min to remove associated food. Animals were then placed in the center of an 8 M glycerol ring marked with xylene cyanol (Sigma-Aldrich X4126). After 10 min, the number of animals within and outside the ring was counted. Data reported are from biologically independent assays performed over multiple days with a minimum of 2 technical replicates per day.

## Microfluidics behavioral assays

Microfluidics behavioral stripe assays using custom designs [56] were performed as previously described [38]. After degassing assembled microfluidic devices, the outlet port was loaded with 5% v/v poloxamer surfactant (Sigma P5556) with 2% xylene cyanol (2 mg/ml) to rid the device of bubbles, and 20 to 30 young adult hermaphrodites were transferred to NGM plates without food flooded with S-basal, then loaded into 2 separated arenas within a single device. After allowing for dispersal (<5 min), the stimulus flow containing the odorant and 2% xylene cyanol (2 mg/ml) for visualization was started; 20-min videos were recorded at 2 Hz on a Pixel-Link camera. Devices were cleaned after each experiment with water, vortexed and soaked in 100% ethanol for at least 24 h, and baked at 50˚C for at least 4 h to evaporate residual ethanol prior to re-use. Mutants were assessed in the same device containing wild-type controls in the adjacent arena.

Custom MATLAB software [56] was used to process and analyze all obtained recordings, and data was visualized using custom R scripts [38] (https://doi.org/10.5281/zenodo.13748735). Mean residency histograms and chemotaxis indices were analyzed as described previously [38]. Each strain was assessed in a minimum of 3 independent assays, performed over multiple days.

## Calcium imaging

Custom microfluidics devices for calcium imaging were fabricated as described previously [38,119]. Images were acquired using an Olympus BX52WI microscope with a 40× oil objective and Hamamatsu Orca CCD camera at 4 Hz with $4 \times 4$ binning. All odorants were diluted in filtered S-basal buffer, and worms were paralyzed in 10 mM (-)-tetramisole hydrochloride (levamisole) (Sigma L9756) prior to loading. Odor dilutions in S-basal buffer were prepared fresh on each day of experiments. Desensitization and adaptation experiments were performed with diacetyl concentrations ($10^{-7}$ corresponding to 1.15 μm diacetyl) that evoke strong desensitization and adaptation [35]. Chemical-evoked calcium transients in ASH were recorded for 2 cycles of 30 s S-basal buffer/30 s odor/30 s S-basal buffer stimulus. For glycerol responses only, data are reported from the second stimulus pulse, as responses to the first glycerol pulse were variable. ASH responses to IAA, quinine, and heptanol are reported from the first pulse. Prior to recording AWA responses, animals were transferred to unseeded plates and starved for 1 h in 10 mM levamisole [35]. To examine AWA responses to odorants including diacetyl, pyrazine, 2-heptanone, and 2-methylpyrazine, odor-evoked calcium transients were recorded for 1 cycle of 30 s buffer/10 s odor/20 s buffer stimulus. In 2 cases (Fig 2B and 2G), a longer stimulus of 30 s odorant was used to fit $t_{1/2}$. Adaptation experiments consisted of a 30 s buffer/10 s odor/20 s buffer stimulus repeated 10 times. Data were collected from biologically independent experiments over at least 2 days. Wild-type controls were imaged on the same day for each experimental condition.

Following image acquisition, neuron fluorescence was first analyzed using FIJI as described previously [38,59]. Briefly, images were aligned using the Template Matching plugin, and cell body and background fluorescence were calculated using manually drawn ROIs. Background-subtracted fluorescence intensity values were used for analysis in RStudio. $F_0$ was calculated as the average $\Delta F/F_0$ value for 5 s prior to odor onset. Rstudio was used to obtain the response mean and SEM (https://doi.org/10.5281/zenodo.13748735).

Peak $\Delta F/F_0$ values were calculated as the maximum change in fluorescence relative to $F_0$ in the 10 s following the addition of odorant. To calculate $t_{1/2}$ for odorant desensitization in AWA, a one phase decay was fit to each trace using Prism 9 software. The time points of maximum fluorescence levels and stimulus removal were designated as the start and end, respectively. Only $t_{1/2}$ values that fit each trace without error are reported in the analysis; the percentage of traces used for each analysis is indicated in the figure legends. Peak $\Delta F/F_0$ values during adaptation experiments were calculated for each pulse, then normalized to the largest response. Peak $\Delta F/F_0$ values across multiple pulses were calculated as the maximum change in fluorescence relative to $F_0$, which was individually calculated for each pulse.

Addition of exogenous GABA and the GABA$_A$ receptor antagonist bicuculline prior to performing odorant-evoked calcium imaging in ASH was performed essentially as described previously [120]. Worms were allowed to crawl for 30 to 60 min on plates containing 50 μm exogenous GABA and ddH$_2$0 control, or 10 μm bicuculline and chloroform control. Plates were also freshly seeded with 50 μl OP50 culture following drug addition. Treated worms were then paralyzed with levamisole and imaged as described above.

## Fluorescent reporter imaging and image analysis

One-day-old young adult hermaphrodites were anesthetized with 10 mM tetramisole hydrochloride (Sigma-Aldrich L9756) and mounted on 10% agarose pads set on microscope slides. Images were acquired at 0.27 μm *z*-intervals using a 100× oil immersion objective on an inverted spinning disk confocal microscope (Zeiss Axiovert with a Yokogawa CSU22 spinning disk confocal head and a Photometerics Quantum SC 512 camera) and Slidebook 6.0

(Intelligent Imaging Innovations, 3i) software. Images of ASH cilia length in Fig 1 were obtained using a 63× oil immersion objective on an upright microscope (Zeiss Imager.M2 with a Hamamatsu C4742.95 camera) and Zeiss Zen software. For all imaging, conditions were kept identical across wild-type and mutant conditions. All image processing and analyses were performed using FIJI/ImageJ (NIH). Data from at least 2 biologically independent experiments performed on independent days were quantified for each data point.

ASH cilia length: ASH cilia length was measured by drawing a line segment from the base of the cilium to the tip. AWA PCMC branches: AWA PCMC branches were categorized as wt or mutant based on the number of segments originating from the PCMC; 0 to 4 branches was considered wild type, and all wild-type controls fell into this category. Five or more branches emerging from anywhere within the PCMC was considered mutant.

Measurement of fluorescence intensities: Fluorescence intensities within AWA cilia were quantified by first manually outlining the cilia area in the maximum projection images. The corrected total cell fluorescence (CTCF) was calculated to account for differences in the extent of cilia branching. Fluorescence intensities at the cilia base were quantified by creating a 1-μm diameter circular ROI at the cilia base in the maximum projection images, then measuring the mean fluorescence intensity within the ROI following background subtraction. Fluorescence intensities within the AWA soma were similarly quantified by manually creating an ROI around the soma and subtracting the background fluorescence from the mean fluorescence value.

Measurement of ODR-10::split-GFP or SRX-64::split-GFP fluorescence within PCMC branches: To determine the percentage of AWA PCMC branches occupied by SRX-64::split-GFP or ODR-10::split-GFP fluorescence, a manually drawn ROI was generated around all PCMC branches. SRX-64::split-GFP or ODR-10::split-GFP fluorescence levels were thresholded within this ROI using a control value acquired from wild-type ODR-10::split-GFP or SRX-64::split-GFP fluorescence images, determined visually from 3 to 5 control images. Percent occupancy was calculated as [area of thresholded fluorescence]/[area of ROI]*100.

EV measurements: ODR-10 or SRX-64::split-GFP positive EVs were identified by generating ROIs of the glial soma (marked with *F16F9.3*p::*mCherry*) and manually identifying puncta within the defined ROI.

## Fluorescence recovery after photobleaching (FRAP)

Fluorescence recovery after photobleaching (FRAP) experiments were conducted using a Zeiss LSM880 with Airyscan using a 63× oil immersion objective. ODR-10::split-GFP fluorescence was photobleached using a 405 nm laser (at 90% power). The entire cilium stalk (approximately 2.5 μm) or a small segment (approximately 2.5 μm) of the PCMC branches was photobleached in wild type or *osm-6* mutants. Five pre-bleach images were obtained, and recovery was imaged for an additional 55 frames (approximately 45 s). Images were normalized to the pre-bleach values and an adjacent reference area of unbleached dendrite using the following double normalization equation [121]:

$$\text{Norm}(t) = \frac{\text{Ref}_{\text{pre-bleach}}}{\text{ref}_{(t)}} * \frac{\text{FRAP}_{(t)}}{\text{FRAP}_{\text{pre-bleach}}}$$

where $\text{Ref}_{\text{pre-bleach}}$ and $\text{FRAP}_{\text{pre-bleach}}$ = mean fluorescence of the reference area and photobleached area pre-bleach, and $\text{ref}_{(t)}$ and $\text{FRAP}_{(t)}$ = the mean fluorescence of the reference area or photobleached area at time t. Images were corrected for drift (Template Matching Plugin; ImageJ), and the Measure Stack macro was used to quantify the mean fluorescence of ROIs.

Norm(t) values were normalized to the max value = 1 and recovery was graphed starting at the minimum fluorescence value within 5 frames of photobleaching.

## IFT Measurements

IFT analyses were performed essentially as described [60,122]. Briefly, videos of OSM-6::GFP or split-GFP movement in ASH or AWA cilia were acquired on a spinning disk confocal microscope for 30 s with 250 ms exposure. IFT in ASH cilia was analyzed by generating a line segment in the region beginning at the transition zone continuing throughout the middle segment. IFT in AWA cilia was analyzed only throughout the proximal ciliary stalk [60]. Kymographs were generated using the Multi Kymograph Plugin (ImageJ) and Template Matching Plugin (ImageJ) to correct for any movement artifacts. IFT particles were identified manually. Line segments over each individual track were drawn to derive velocity measurements.

## Dendritic trafficking

Movement of ODR-10::split-GFP puncta within AWA dendrites was visualized using the Multi Kymograph Plugin (ImageJ) and Template Matching Plugin (ImageJ) to correct for any movement artifacts. Videos were acquired on a spinning disk confocal microscope for 30 s with 250 ms exposure, and puncta within dendrites were visualized manually.

## Statistical analyses

All graphs were generated and analyzed using RStudio and Prism 9 software. Scatterplots and shading throughout figures indicate mean ± SEM. For comparisons between 2 groups, mutant and control genotypes were compared using an unpaired *t* test with equal SD. For comparisons among multiple strains, analyses used a one-way ANOVA followed by Tukey's multiple comparisons test. Adaptation analysis used a second-way ANOVA with repeated measures. Fisher's exact test was used to compare IFT movements, comparing different IFT categories between 2 groups. Further details of the numbers of animals analyzed and significance values for each test are provided in each figure legend.

## Supporting information

**S1 Fig. Mutations in *osm-6* abolish responses to aqueous but not volatile chemicals in ASH. (A, B, D, E)** Mean GCaMP3 fluorescence changes (top, A, B; left, D, E) and quantification of peak fluorescence intensities (bottom, A, B; right, D, E) in ASH to a 30 s pulse of the indicated chemicals in wild-type and *osm-6(p811)* mutants. Shaded regions indicate SEM. $n \geq 15$ neurons each. ** and ***: different from wild type at $P < 0.01$ and $P < 0.001$, respectively (*t* test); ns: not significant. Animals were treated with either 50 μm GABA or 10 μm biculline compared to their corresponding controls in **E**. For glycerol responses only, data are reported from the second stimulus pulse (see Methods); the 0 time point represents 90 s following initiation of imaging. **(C)** Desensitization rates for the IAA response in ASH shown in Fig 1B in wild-type or *osm-6* mutants; >92% of traces fit $t_{1/2}$. *: different from wild type at $P < 0.05$ (*t* test). Each dot in the scatter plots is the measurement from a single ASH neuron. Horizontal lines in all plots indicate the mean. Errors are SEM. Data shown are from a minimum of 2 independent experiments. Underlying data are provided in https://doi.org/10.5281/zenodo.13748735.
(EPS)

**S2 Fig. Characterization of the *osm-3(ts)* and endogenously tagged *osm-6* alleles. (A)** Sequence alignment of *C. elegans* kinesin motor subunits OSM-3, KLP-11, and KLP-20 with

the *Chlamydomonas* kinesin motor subunits FLA10 and FLA8. The phenylalanine residue mutated to serine in FLA8-2 and engineered in OSM-3 to create *osm-3(oy156ts)* is indicated. **(B)** Percentage of animals of the indicated genotypes exhibiting defects in DiI uptake (Dyf) at 20˚C or upon temperature upshift to 30˚C for 4 h. $n$ = 3 assays with 100 animals per genotype/ condition. **(C)** ASH cilia length (left) or dye-filling (right) in animals carrying the *osm-6::gfp$_{11}$* allele (*oy166*). Each dot is the value from a single neuron. Horizontal lines indicate the mean. Errors are SEM. ns: not significant. $n$ = 3 assays, 100 animals per genotype (dye-filling). **(D)** Histogram of velocities of animals carrying the *osm-6::gfp$_{11}$ (oy166)* allele and *gfp$_{1-10}$* expressed under the *sra-6* promoter in an extrachromosomal array, or expressing *sra*-6p::OSM-6::GFP from an extrachromosomal array. $n \geq 20$ animals per genotype. Underlying data are provided in https://doi.org/10.5281/zenodo.13748735.
(EPS)

**S3 Fig. AWA retains responses, but exhibits desensitization defects, to a subset of volatile odorants in *osm-6* mutants. (A, B)** Mean GCaMP2.2b fluorescence changes (top) and quantification of peak fluorescence intensities and desensitization rates (bottom) in AWA to a 10 s pulse of the indicated odorants in animals of the shown genotypes. Shaded regions indicate SEM. $n \geq 21$ neurons each. *, ** and ***: different from corresponding wild type at $P < 0.01$, $P < 0.05$, and $P < 0.001$, respectively ($t$ test). ns: not significant. Fewer than 25% of traces fit $t_{1/2}$ in (A); thus, 30 s pulse data were analyzed instead (Fig 2B). At least 68% and 48% of traces fit $t_{1/2}$ for $10^{-4}$ and $10^{-5}$ 2-heptanone, respectively (B). **(C, D)** Mean GCaMP2.2b fluorescence changes (left) and quantification of peak fluorescence intensities (right) in AWA to a 10 s pulse of the indicated odorants in animals of the shown genotypes. Shaded regions indicate SEM. $n \geq 16$ neurons each. *and **: different from corresponding wild type at $P < 0.05$ and $P < 0.001$, respectively ($t$ test). **(E)** Chemotaxis indices of animals of the indicated genotypes to the shown concentrations of odorants. Each dot is the chemotaxis index of a single assay, containing 100–300 adult hermaphrodites, $n \geq 5$ assays. ** and ***: different at $P < 0.01$ and $P < 0.001$ from corresponding wild type (one-way ANOVA with Tukey's multiple comparisons test). Dia–diacetyl; Pyr–pyrazine. Each dot in the scatter plots is the measurement from a single AWA neuron. Horizontal lines in all plots indicate the mean. Errors are SEM. Data shown are from a minimum of 2 independent experiments. All calcium imaging was initiated 30 s prior to stimulus onset; responses from 5 s prior to stimulus onset are shown. Underlying data are provided in https://doi.org/10.5281/zenodo.13748735.
(EPS)

**S4 Fig. Analysis of chemotaxis behaviors and odorant receptor expression in IFT mutants. (A)** Chemotaxis indices of animals of the indicated genotypes to pyrazine. Each dot is the chemotaxis index of a single assay, containing 100–300 adult hermaphrodites, $n \geq 5$ assays. ns: not significant. *srx-64::gfp$_{11}$* is allele *srx-64(oy193)*, *gfp$_{1-10}$* is expressed under the *gpa-4Δ6* promoter. **(B)** Representative image of endogenously tagged SRX-64::split-GFP expression in AWA cilia in wild-type or *osm-5(p813)* mutants. Numbers at bottom right indicate the percentage of animals exhibiting the shown phenotype; $n \geq 20$. White/yellow arrowheads indicate cilia tip/base. Scale bar: 5 μm. **(C)** Representative images (top) and quantification (bottom) of SRX-64::SL2::split-GFP fluorescence in the AWA soma in the indicated genotypes. Scale bar: 5 μm. Each dot is the value from a single AWA neuron. ***: different at $P < 0.001$ from wild type ($t$ test). Horizontal lines in all plots indicate the mean. Errors are SEM. Data shown are from a minimum of 2 to 3 independent experiments. Underlying data are provided in https://doi.org/10.5281/zenodo.13748735.
(EPS)

**S5 Fig. A subset of signaling molecules is localized to the PCMC branches in AWA. (A)** Representative images of AWA sensory endings in *kap-1; osm-3(ts)* mutants prior to and following temperature shift to the restrictive temperature for 24 h. Numbers at bottom right indicate the percentage of animals exhibiting PCMC branches; $n \geq 26$. **(B–D)** Representative images of indicated reporter-tagged proteins in wild-type and *osm-6(p811)* mutants. Numbers at bottom right indicate the percentage of neurons exhibiting the shown phenotype; $n \geq 20$ each. White/yellow arrows indicate cilia tip/base. Arrows indicate PCMC branches. Scale bars: 5 μm.
(EPS)

**S6 Fig. *grk-2* shapes odorant response amplitudes in AWA. (A)** Quantification of peak fluorescence intensity changes for data shown in Fig 4A and 4B. * and ***: different at $P < 0.05$ and $P < 0.001$ from indicated genotype (one-way ANOVA with Tukey's multiple comparisons test–left, *t* test–right). ns: not significant. **(B)** Representative images of AWA cilia in wild-type and *grk-2(gk268)* animals expressing *gpa-4Δ6p::myr-GFP*. White/yellow arrows indicate cilia tip/base. Numbers at top right indicate percentage of animals exhibiting the shown phenotype; $n \geq 20$ each. Scale bar: 5 μm. Underlying data are provided in https://doi.org/10.5281/zenodo.13748735.
(EPS)

**S7 Fig. Diacetyl adaptation in AWA requires the GRK-2 kinase and IFT. (A)** Adaptation of peak AWA fluorescence upon repeated 10 s pulses of diacetyl in the indicated genotypes, normalized to the maximum peak $\Delta F/F_0$. $n \geq 14$ animals for each data point. ** and ***: different at $P < 0.01$ and $P < 0.001$ from wild type (second-way ANOVA with repeated measures). **(B)** Percentage recovery of ODR-10::split-GFP fluorescence 40 s after photobleaching the AWA ciliary stalk (red) or a similarly sized region of PCMC branches (black) in *osm-6(p802)* mutants. Data in red are also shown in Fig 5B. **: different at $P < 0.01$ from indicated (*t* test). **(C)** Representative images (left) and quantification (right) of ODR-10::split-GFP levels in wild-type and *grk-2(gk268)* animals. Wild-type *grk-2* sequences tagged with tagRFP were expressed in AWA under the *gpa-4Δ6* promoter. ** and ***: different at $P < 0.01$ and $P < 0.001$ between indicated values (one-way ANOVA with Tukey's multiple comparisons test). **(D)** A predicted BBSome-binding motif ([W/F/Y]R) is located within helix 8 of ciliary GPCRs [80] but is absent in SRX-64. Conserved BBSome-binding residues are shown in bold. **(E)** Representative images of animals expressing wildtype ODR-10::split-GFP or a mutant protein containing a FR-to-AA substitution in the predicted BBSome binding site shown in D at the endogenous locus. Numbers at bottom right indicate percentage of animals exhibiting the shown phenotype; $n \geq 20$ each. **(F)** Adaptation of peak AWA fluorescence upon repeated 10 s pulses of diacetyl in the indicated genotypes, normalized to the maximum peak $\Delta F/F_0$. $n \geq 9$ animals for each data point. (Left) *,** and ***: different at $P < 0.05$, $P < 0.01$, and $P < 0.001$ from wild type (second way ANOVA with repeated measures), (right) ns: not significantly different from *grk-2* (second way ANOVA with repeated measures). **(G)** Summary of hypothesized IFT-dependent and independent removal of ODR-10 and SRX-64 GPCRs, respectively, during odorant adaptation. Upon interaction with diacetyl, ODR-10 is phosphorylated by GRK-2 and removed from cilia via BBSome and IFT. ODR-10 is then subsequently trafficked retrogradely to the AWA soma. In contrast, upon pyrazine-mediated activation of SRX-64, this receptor may be removed from cilia via ectocytosis. In all images, yellow/white arrowheads indicate cilia base/cilia tip. Each dot in scatter plots is the measurement from a single AWA neuron. Horizontal lines indicate the mean. Errors are SEM. Data shown are from a minimum of 2 to 3 independent experiments. ns: not significant. Scale bars: 5 μm. Underlying data are

**S1 Movie. Movement of endogenously tagged OSM-6::split-GFP within the primary AWA ciliary stalk in wildtype animals.** Arrowhead marks cilia base, distal end of cilia is at top. Images were captured at 4 Hz and played back at 20 frames per sec.
(MP4)

**S2 Movie. Movement of endogenously tagged EBP-2::split-GFP movement within dendritic branches in *osm-6(p811)* mutants.** Arrowhead marks PCMC branches, distal end of cilia is at top. Images were captured at 4 Hz and played back at 20 frames per sec.
(MP4)

**S3 Movie. Trafficking of endogenously tagged ODR-10::split-GFP puncta in the AWA dendrite upon addition of $10^{-4}$ dilution of diacetyl.** Cilia at left, soma at right. Images were captured at 4 Hz and played back at 20 frames per sec.
(MP4)

**S4 Movie. Trafficking of endogenously tagged SRX-64::split-GFP puncta in the AWA dendrite upon addition of $10^{-4}$ dilution of pyrazine.** Cilia at left, soma at right. Images were captured at 4 Hz and played back at 20 frames per sec.
(MP4)

**S1 File. List of strains used in this work.**
(DOCX)

**S2 File. List of plasmids used in this work.**
(DOCX)

**S1 Table. Quantification of OSM-6::GFP or OSM-6::split-GFP anterograde movement in ASH cilia.**
(DOCX)

**S2 Table. Quantification of OSM-6::split-GFP anterograde movement in the AWA cilia stalk.**
(DOCX)

## Acknowledgments

We are grateful to Kirsten Judge and Ashish Maurya for providing reagents, the *Caenorhabditis* Genetics Center and the National BioResource Project (Japan) for providing strains, Ed Dougherty and Andy Stone in the Brandeis light microscopy core facility for assistance with imaging and image analyses, Frank Mello in the Brandeis Machine shop for technical troubleshooting, the Brandeis Materials Research Science and Engineering Center (MRSEC) for access to the microfabrication facility, and Dirk Albrecht for assistance with chip design and behavioral analyses. We thank members of the Sengupta lab for experimental advice, and members of the Sengupta lab, Max Heiman, Max Nachury, and Cori Bargmann for critical comments on the manuscript.

## Author Contributions

**Conceptualization:** Alison Philbrook, Piali Sengupta.

**Data curation:** Alison Philbrook.

**Formal analysis:** Alison Philbrook, Michael P. O'Donnell.

**Funding acquisition:** Piali Sengupta.

**Investigation:** Alison Philbrook.

**Methodology:** Alison Philbrook, Michael P. O'Donnell.

**Project administration:** Piali Sengupta.

**Resources:** Alison Philbrook, Laura Grunenkovaite.

**Software:** Michael P. O'Donnell.

**Supervision:** Piali Sengupta.

**Validation:** Alison Philbrook.

**Visualization:** Alison Philbrook, Laura Grunenkovaite.

**Writing – original draft:** Alison Philbrook, Piali Sengupta.

**Writing – review & editing:** Alison Philbrook, Michael P. O'Donnell, Laura Grunenkovaite, Piali Sengupta.

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
