## [Editor Report · Decision Letter 0]

21 May 2024

Dear Piali, 

Thank you for submitting your manuscript entitled "Differential modulation of sensory response dynamics by cilia structure and intraflagellar transport within and across chemosensory neurons" for consideration as a Research Article by PLOS Biology.

Your manuscript has now been evaluated by the PLOS Biology editorial staff as well as by an academic editor with relevant expertise and I am writing to let you know that we would like to send your submission out for external peer review.

Once your full submission is complete, your paper will undergo a series of checks in preparation for peer review. After your manuscript has passed the checks it will be sent out for review. To provide the metadata for your submission, please Login to Editorial Manager (https://www.editorialmanager.com/pbiology) within two working days, i.e. by May 23 2024 11:59PM.

Kind regards,

Ines

--

Ines Alvarez-Garcia, PhD

Senior Editor

PLOS Biology

---

## [Decision Letter · Decision Letter 1]

24 Jul 2024

Dear Piali,

Thank you for your patience while your manuscript entitled "Differential modulation of sensory response dynamics by cilia structure and intraflagellar transport within and across chemosensory neurons" was peer-reviewed at PLOS Biology. Please also accept my sincere apologies again for the delay in sending you our decision. The manuscript has now been evaluated by the PLOS Biology editors, an Academic Editor with relevant expertise, and we consulted three independent reviewers, although we have to date only received reports from two of them; we will forward you the third one if it is sent to us belatedly.

As you will see, the reviewers find the conclusions interesting, but they also raise several concerns that would need to be addressed to strengthen the results. Reviewer 1 thinks that by analysing only two types of neurons, it is difficult to determine if the observed differences reflect true mechanistic heterogeneity across several neuronal types or are specific to the differences between ASH and AWA neurons. In addition, the chemical cues are limited to two compounds per neuron, each tested at a single concentration, thus the reviewer thinks that dose-response experiments should be performed for all components, the number of chemicals tested per receptor type should be increased and analyse one additional type of nociceptive neuron and one additional type of olfactory neuron at least, to clearly ascribe different signalling properties to cilia differences rather than chemical or other neuronal differences. Reviewer 2 asks for several clarifications on the way IFT might respond to chemicals, test if changes in AMsh glia- ASH GABAergic signalling are involved in the response, and if cilia structure and IFT could act in a similar way in the regulation of AWA responses to low and high concentrations of diacetyl.

In light of the reviews and after consulting with the Academic Editor, we would like to invite you to revise the work to thoroughly address the reviewers' reports. Given the extent of revision needed, we cannot make a decision about publication until we have seen the revised manuscript and your response to the reviewers' comments. Your revised manuscript is likely to be sent for further evaluation by all or a subset of the reviewers.

**IMPORTANT - SUBMITTING YOUR REVISION**

3. Resubmission Checklist

a) *PLOS Data Policy*

b) *Published Peer Review*

d) *Blurb*

Please also provide a blurb which (if accepted) will be included in our weekly and monthly Electronic Table of Contents, sent out to readers of PLOS Biology, and may be used to promote your article in social media. The blurb should be about 30-40 words long and is subject to editorial changes. It should, without exaggeration, entice people to read your manuscript. It should not be redundant with the title and should not contain acronyms or abbreviations. For examples, view our author guidelines: https://journals.plos.org/plosbiology/s/revising-your-manuscript#loc-blurb

Sincerely,

Ines

--

Ines Alvarez-Garcia, PhD

Senior Editor

PLOS Biology

Reviewers' comments

Rev. 1:

In the study titled "Differential modulation of sensory response dynamics by cilia structure and intraflagellar transport within and across chemosensory neurons," Philbrook et al. investigate how intraflagellar transport (IFT) and cilia structure differentially contribute to the responses of two types of chemosensory neurons: ASH and AWA. In ASH nociceptive neurons, the authors discover that a minimum cilia length is essential for responses to 1 M glycerol, but not to isoamyl alcohol (IAA) at 10-2 dilution. Conversely, in AWA olfactory sensory neurons, neither cilia length nor continuous IFT affect the peak response to diacetyl at a 10-7 dilution, but both factors significantly reduce the response to pyrazine at a 10-6 dilution. Moreover, the authors discover that IFT is crucial for the adaptation of AWA neurons to both odorants. Mechanistically, the different phenotypes in response to diacetyl and pyrazine are due to distinct transcriptional regulation, subcellular localization, and receptor internalization or exocytosis of the odorant receptors, ODR-10 and SRX-64, in IFT mutants.

This in-depth genetic study reveals how IFT and cilia length differentially regulate chemosensory responses within and between different types of neurons. The genetic dissection of how ODR-10 and SRX-64 are differentially affected by IFT impairments in the AWA neuron is an example of rigorous experimental design, showcasing the powerful genetic tools available in C. elegans research. The application of acute inhibition of IFT without disrupting cilia morphology is particularly elegant. This study thus has the potential to advance the field of cellular sensory neurobiology. While I have one major concern and a few specific comments, addressing these points will enhance the overall clarity and impact of the study.

Major concern:

The study claims to have uncovered surprising complexity in the diverse contribution of IFT and cilia morphology to the regulation of responses across individual chemosensory neurons to diverse chemical cues. However, the authors only examine two types of neurons: the ASH nociceptive neuron and the AWA olfactory neuron. Therefore, it is difficult to determine whether the observed differences reflect true mechanistic heterogeneity across various neuronal types or are specific to the differences between ASH and AWA neurons. In addition, the "diverse chemical cues" are limited to two compounds for each neuron, each tested at a single concentration. Although the preliminary data appear promising, it remains uncertain whether the differences in response phenotypes are due to specific chemical structures or differences in chemical concentration. To address these concerns, it is essential to conduct dose-response experiments (for all compounds shown in Fig 1B, Figs 2B and 2C). Additionally, the authors need to increase the number of chemicals tested per receptor type, and examine at least one additional type of nociceptive neuron and one additional type of olfactory neuron in their initial functional characterization (as in Figs 1 and 2).

Specific comments:

(1) Fig 1B: The IAA response kinetics also appear to be affected in osm-6, showing a faster decay. Please provide a proper kinetics analysis across various IAA concentrations.

(2) Fig 1B: It appeared that IAA was presented before glycerol. Please tested the reverse sequence to determine whether the order of stimulus presentation affects the phenotype.

(3) Fig 1C: Please perform a proper correlation analysis between the cilium length and response amplitude to support the claim made in Page 6 ("we correlated cilium length with glycerol-evoked Ca2+ dynamics….").

(4) Fig 1H: Please also test IAA and other chemicals.

(5) If the responses of IAA to glycerol and IAA are indeed differentially regulated by cilia morphology, what could be the underlying molecular mechanism? It is necessary to test additional aqueous and volatile odorants before any generalization can be made.

(6) Fig 2B: Is the adaptation phenotype observed only with 30-sec stimulation? How about 10 sec?

(7) Fig 2C: Why did the authors change the stimulus duration from 30 sec to 10 sec? Please also provide a kinetics analysis here.

(8) Fig 3A: change the 1.5 hr shift/plus sign to 30⁰C.

(9) Fig 3C: Why did the authors change the stimulus duration from 30 sec to 10 sec? Please also analyze the response kinetics across concentrations.

(10) Fig 4C & D: The authors elegantly demonstrated the lack of impact of IFT on nociceptive response and behavior in Fig 1J-K. Please conduct behavioral assays using the inducible mutants (5-hr shift) in a similar manner to demonstrate the specific impact of IFT on olfactory behavior.

(11) Fig 5E, right panels: It is difficult to visualize EVs in the image.

(12) Page 26: The authors stated "these observations support the notion that phosphorylated ODR-10 may be removed from AWA cilia via BBSome-mediated trafficking to mediate diacetyl adaptation". This notion can be strengthened with epistasis experiments to test whether Grk2 phosphorylation of ODR-10 is upstream of BBSome-mediated trafficking.

(13) Methods: Please specify the age of all experimental animals as the manifestation of phenotypes may be age-dependent.

Rev. 2:

Philbrook et al. explored the contributions of ciliary trafficking (IFT) and cilia structure to chemosensory responses in the sensory neurons of C. elegans. By selectively inhibiting IFT without altering cilia structure, and vice versa, they demonstrated that for ASH nociceptive neurons, a minimum cilium length is necessary for the responses to a subset of chemicals such as glycerol, but continuous IFT is not. In contrast, neither cilia nor continuous IFT are required for odorant responses in AWA neurons. Instead, continuous IFT influences response dynamics in AWA, where inhibiting IFT causes odorant receptors to be misrouted, leading to desensitization defects. Additionally, they discovered that the adaptation of AWA responses to repeated exposure to diacetyl is mediated by IFT-driven receptor removal, while adaptation to a different odorant pyrazine occurs via IFT-independent mechanisms. These findings reveal the unexpected complexity in the roles of IFT and cilia organization in regulating chemosensory responses both within and across chemosensory neurons, underscoring their essential roles in the precise modulation of olfactory behaviors. The basic conclusions of this study are well supported. However, there are certain aspects of the manuscript that would benefit from further clarification.

The authors infer that "a minimum cilium length may be necessary for glycerol but not isoamyl alcohol responses in ASH. Alternatively, IFT may differentially regulate responses to these chemicals in ASH". Does IFT affect responses to IAA in ASH? Since isoamyl alcohol is a volatile odorant, it may directly penetrate the worm's cuticle and affect the short cilia, dendrite, or even the cell body. Is it possible that glycerol is not a volatile odorant, so it can only affect the cilia of ASH through the opening channel, thus requiring a longer cilium?

Previous studies (Duan et al.,Neuron 2020; Cheng et al., Neuron 2024) have reported that ASH adaptation is mediated in part via GABA release from the AMsh glia. Is it possible that mutations disrupting cilia structure affect glycerol- and/or IAA- responses via changes in AMsh glia- ASH GABAergic signalling?

Page 22: "While neurons with simple rod-like cilia such as ASE respond to salts over only a 4-fold concentration range, AWA detects diacetyl over a 10^5-fold range." It has been suggested (Taniguchi et al., Science Signaling 2014) that for diacetyl reception, ODR-10 in AWA neurons mediates the attractive response to low concentrations, while SRI-14 in ASH neurons mediates the avoidance response to high concentrations. Is the role and mechanism of cilia structure and IFT similar in regulating AWA responses to both low and high concentrations of diacetyl?

---

## [Decision Letter · Decision Letter 2]

30 Sep 2024

Dear Piali,

Thank you for your patience while we considered your revised manuscript entitled "Differential modulation of sensory response dynamics by cilia structure and intraflagellar transport within and across chemosensory neurons" for publication as a Research Article at PLOS Biology. This revised version of your manuscript has been evaluated by the PLOS Biology editors, the Academic Editor and the two original reviewers.

Based on the reviews, we are likely to accept this manuscript for publication, provided you satisfactorily address the remaining points raised by Reviewer 1. Please also make sure to address the data and other policy-related requests stated below.

In addition, we would like you to consider a suggestion to improve the title, taking also into consideration Rev. 1's request.

"Cilia structure and intraflagellar transport regulate sensory response dynamics within and between C. elegans chemosensory neurons"

We expect to receive your revised manuscript within two weeks. 

*Published Peer Review History*

*Press*

Sincerely,

Ines

--

Ines Alvarez-Garcia, PhD

Senior Editor

PLOS Biology

DATA POLICY:

Thank you for providing the data underlying the graphs shown in the figures in Zenodo. I had a look and find it hard to match the data from each figure to each graph and it would be very useful if you could add a list matching the data of each graph to the corresponding files. We do require the data underlying the graphs shown in the following figures:

Fig. 1A-K; Fig. 2B-D, G; Fig. 3A, C-E; Fig. 4A-E; Fig. 5A-E; Fig. S1A-E; Fig. S2B-D; Fig. S3A-E; Fig. S4A, C; Fig. S6A and Fig. S7A-C, F

Reviewers' comments

Rev. 1:

The authors' efforts in addressing the reviewers' comments are commendable. I only have two editorial suggestions to further improve the manuscript:

1. Title modification: Change "across chemosensory neurons" to "between chemosensory neurons" in the title. This change more accurately reflects the study's focus on two specific types of neurons.

2. Time series plot revisions: To enhance clarity in Figure 1B and other panels containing time series plots, I recommend the following adjustments: a) Standardize the X-axis: Make all panels start at 0 sec to maintain consistency and avoid confusion. For example, in Figure 1B, the left panel starts at 90 sec while the right panel starts at 0 sec, which gives readers the false impression that IAA was presented before glycerol. b) Define the zero time point: Indicate that time zero represents xx seconds prior to stimulus onset.

Rev. 2: Lijun Kang - note that this reviewer has signed his review.

The authors have thoughtfully responded to my suggestions, and several new experiments have strengthened the paper. I think it is now suitable for publication in BLOS Biology.

---

## [Editor Report · Decision Letter 3]

10 Oct 2024

Dear Piali,

Thank you for the submission of your revised Research Article entitled "Cilia structure and intraflagellar transport differentially regulate sensory response dynamics within and between C. elegans chemosensory neurons" for publication in PLOS Biology. On behalf of my colleagues and the Academic Editor, Richard Benton, I am delighted to let you know that we can in principle accept your manuscript for publication, provided you address any remaining formatting and reporting issues. These will be detailed in an email you should receive within 2-3 business days from our colleagues in the journal operations team; no action is required from you until then. Please note that we will not be able to formally accept your manuscript and schedule it for publication until you have completed any requested changes.

PRESS

Sincerely, 

Ines

--

Ines Alvarez-Garcia, PhD

Senior Editor

PLOS Biology
